# Robust Sequential Experimental Design for A/B Testing

**Qianglin Wen** [* 1]  **Xiangkun Wu** [* 2]  **Chengchun Shi** [3]  **Ting Li** [4]  **Niansheng Tang** [1]  **Yingying Zhang** [† 5]  **Hongtu Zhu** [† 6]

## Abstract

Experimental design has emerged as a powerful approach for improving the sample efficiency of A/B testing, yet existing designs rely critically on correctly specified models. We study robust sequential experimental design under model misspecification and develop a unified framework that covers both contextual bandit and dynamic settings. Theoretically, we prove that our design bounds the worst-case mean squared error of the estimated treatment effect. Empirically, we demonstrate the effectiveness of the proposed approach using synthetic and real-world datasets from a leading technology company.

## 1. Introduction

A/B testing plays a crucial role in modern technology companies for data-driven decision making and product deployment (Johari et al., 2017; Shi et al., 2021; Waudby-Smith et al., 2024; Johari et al., 2025). In its simplest form, it randomly and independently allocates each experimental unit to either the treatment or the control group, and then compares their mean outcomes to estimate the average treatment effect (ATE, Imbens & Rubin, 2015).

While simple, independent random assignment can be highly inefficient in practice. Although treatment and control groups share the same covariate distributions at the population level, finite-sample randomness often leads to sub-

---

[*]Equal contribution  [1]Yunnan Key Laboratory of Statistical Modeling and Data Analysis, Yunnan University, Kunming, China [2]School of Mathematical Sciences, Zhejiang University, Hangzhou, China [3]Department of Statistics, London School of Economics and Political Science, London, United Kingdom [4]School of Statistics and Data Science, Shanghai University of Finance and Economics, Shanghai, China [5]School of Statistics, East China Normal University, Shanghai, China [6]University of North Carolina at Chapel Hill, Chapel Hill, NC, United States. Correspondence to: Yingying Zhang <yyzhang@fem.ecnu.edu.cn>, Hongtu Zhu <htzhu@email.unc.edu>.

*Proceedings of the 43^{rd} International Conference on Machine Learning*, Seoul, South Korea. PMLR 306, 2026. Copyright 2026 by the author(s).

stantial imbalances in realized samples. Consequently, differences in outcomes between treatment and control groups may be driven by such imbalances rather than true treatment effects (Masoero et al., 2023). Such covariate imbalances can commonly occur as A/B testing is often conducted over a limited time horizon, which constrains the effective sample size for learning (Athey et al., 2023; Bojinov et al., 2023; Shi et al., 2023b). Moreover, treatment effects are typically small, making them difficult to detect with limited data (Tang et al., 2019; Farias et al., 2022; 2023; Xiong et al., 2024; Li et al., 2024b; Wang et al., 2025; Wu et al., 2025). These challenges have motivated a growing literature on sequential experimental designs that maintain covariate balance. Such designs are tailored to sequential settings where experimental units arrive over time, requiring assignments to be made upon arrival and without knowledge of future units. Section 2 offers a more detailed survey of this literature.

Despite their widespread use, these sequential designs are inherently *myopic*: they focus on minimizing immediate imbalance measures without accounting for how current assignments influence the state of future covariates (Bhat et al., 2020). While subsequent designs have optimized long-term objectives (see Section 2), they require correctly specified linear models, additive treatment effects and certain covariate distributional assumptions. Furthermore, they assume the absence of carryover effects (where past treatments influence future outcomes), despite the prevalence of such effects in sequential settings (Robins, 1986; Han et al., 2022; Viviano & Bradic, 2023; Xu et al., 2023; Zhang et al., 2023; Li et al., 2024a; Shi et al., 2023a; 2024; Chen & Simchi-Levi, 2025).

This paper studies *robust sequential designs* for A/B testing in a misspecified environment, where the ATE is estimated using a linear working model while the true data-generating process may contain unknown nonlinear components. Our proposal has three ingredients:

(i) An *orthogonalization* technique that yields a robust upper bound on the mean squared error (MSE) of the ATE estimator without requiring knowledge of the specific nonlinear effects (Section 3.2);

(ii) A *dynamic programming* (DP) algorithm that enables sequential allocation while optimizing a long-term objective, namely the established MSE upper bound (Section 3.3);

(iii) An extension of our methodology to dynamic settings with carryover effects, with a *hierarchical design* that integrates DP and reinforcement learning (RL) to facilitate efficient optimization of the design (Section 4).

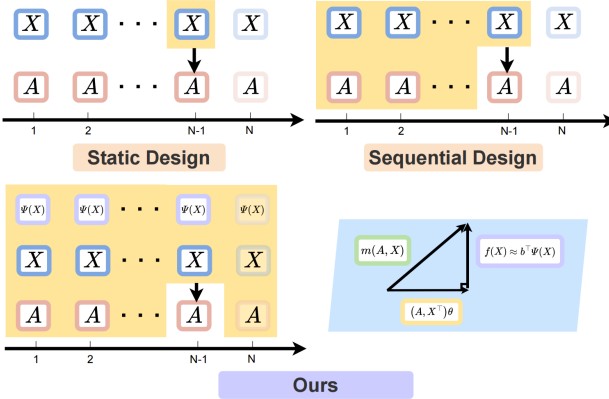

*Figure 1.* Graphical illustration of treatment allocation strategies under different experimental designs. *Static designs* are offline and depend only on current observations. *Sequential designs* condition treatment allocation on the observed history. In contrast, our *robust sequential design* accounts for how current actions affect future covariates, while remaining robust to model misspecification and finite-sample-aware.

As a result, our approach offers three advantages compared to existing designs; see also Figure 1 for a visualization of differences between our design and existing ones. First, it is **robust**, as it explicitly accounts for potential model misspecification through orthogonalization. Second, it is **finite-sample-aware**; rather than relying on large-sample approximations, the assignment rule depends on the entire data history, allowing it to directly mitigate finite-sample randomness in the covariate distributions. Finally, it is **versatile**, since it provides a unified framework that covers both the contextual bandit setting and dynamic environments with carryover effects.

**Conflict of Interest Disclosure.** The authors declare no financial conflicts of interest related to this work.

## 2. Related Works

A/B testing has been extensively studied across multiple disciplines, including statistics, machine learning, operations research and management science; we refer readers to Larsen et al. (2024) and Quin et al. (2024) for two recent comprehensive reviews. While much of the current research

concentrates on estimating the ATE and computing $p$-values for the subsequent hypothesis testing using observed experimental data, this paper shifts the focus to a different and increasingly critical problem:

*How to design and generate the experimental data itself in order to improve the precision of ATE estimation?*

We review three related strands of literature on experimental design below, including (i) methodologies specifically tailored for A/B testing, (ii) sequential experimental designs and (iii) robust experimental designs.

**Experimental design for A/B testing.** Recent studies in management science, machine learning and statistics have increasingly studied A/B testing through the lens of experimental design (Liu et al., 2024; Bajari et al., 2021; 2023; Wang et al., 2023; Kato et al., 2024; Yang et al., 2024; Zhu et al., 2025; Gao et al., 2026; Masoero et al., 2026).

A highly relevant branch of this research focuses on time series experiments involving carryover effects. Their setup closely aligns with the sequential setting of our work where policies are repeatedly assigned over time (Glynn et al., 2020; Hu & Wager, 2022; Basse et al., 2023; Bojinov et al., 2023; Li et al., 2023; Ni et al., 2023; Sun et al., 2024; Xiong et al., 2024; Chen & Simchi-Levi, 2025; Jia et al., 2025; Ni & Bojinov, 2025; Wen et al., 2025; Wu et al., 2026). Experimental designs proposed in these studies can be broadly categorized into three classes: (i) *static designs*, which extend classical Neyman allocation to time series experiments by keeping policy assignments constant within days but varying them across days (Li et al., 2023); (ii) *switchback designs*, which alternate between the new and old policies according to a predetermined schedule (Bojinov et al., 2023; Xiong et al., 2024; Wen et al., 2025); (iii) *short-memory designs*, developed under specific time-series models such as ARMA, where assignment policies depend only on a limited recent history (Sun et al., 2024; Ni & Bojinov, 2025). A special case is *Markov designs*, in which policy assignments are conditioned solely on the current observation rather than the full history (Glynn et al., 2020; Hu & Wager, 2022; Jia et al., 2025).

While these designs effectively capture temporal dynamics and carryover effects, they suffer from two limitations. Methodologically, they rely critically on the correct specification of the underlying time series model, such as a Markov decision process (MDP, Glynn et al., 2020; Hu & Wager, 2022; Li et al., 2023; Wen et al., 2025; Jia et al., 2025) or an ARMA process (Sun et al., 2024). Theoretically, their performance guarantees are mostly asymptotic (see e.g., Li et al., 2023; Sun et al., 2024; Wu et al., 2026), failing to account for the impact of finite-sample randomness. In contrast, our approach is more practically applicable by allowing for model misspecification and strategically allo-

cating treatments using the entire data history to mitigate finite-sample randomness.

**Sequential experimental design**. Sequential experimental design studies the *online allocation problem* where subjects arrive sequentially in order and must be assigned to the treatment or control policy in real time. A classical solution is the *covariate-adaptive biased coin design*, which seeks to balance not only the number of subjects assigned to the treatment and control groups, but also the distribution of observed covariates between them (Pocock & Simon, 1975; Atkinson, 1982; Smith, 1984; Ball et al., 1993; Hu et al., 2009; Baldi Antognini & Zagoraiou, 2011; Hu et al., 2014; 2015; Zhu et al., 2020; Liu et al., 2024; Ma et al., 2024). Such designs typically determine assignment probabilities by minimizing a loss function that quantifies the current degree of covariate imbalance between the two groups. Other works, such as Kapelner & Krieger (2014), propose heuristic approaches based on online matching to further improve the covariate balance. However, as discussed in the introduction, these designs are myopic.

To move beyond myopic designs, recent work has studied sequential experimental design through (approximate) DP (Huan & Marzouk, 2016). In the context of A/B testing, Bhat et al. (2020) make a pioneering attempt to apply DP to treatment allocation and optimize long-horizon objectives. Specifically, they use the MSE of the ATE estimator as the ultimate objective function, apply Bellman equations to derive intermediate objectives at each time point, obtain tractable solutions to solve these intermediate objectives under certain covariate distributional assumptions, and offer rigorous finite-sample performance guarantees of their design. Their proposal is fundamentally different from traditional covariate-imbalance–based myopic designs.

However, their approach relies on strong modeling assumptions, including a correctly specified linear outcome model with additive treatment effects and elliptical covariate distributions. These assumptions are frequently violated in practice: outcomes often exhibit nonlinear dependence on covariates, treatment effects may vary with covariates, and covariate distributions can be substantially more complex. Moreover, their work assumes the absence of carryover effects. As a result, despite the conceptual appeal of dynamic programming for experimental design, relatively few works have extended this approach beyond such restrictive settings. Our work substantially advances this line of research by relaxing these restrictive modeling assumptions and accommodating carryover effects.

**Robust experimental design.** Robust experimental design is a classical line of work in statistics, where design points are chosen to hedge against model misspecification (Fang & Wiens, 2000; Wiens, 2000; Shi et al., 2007; Wiens, 2009; Kong & Wiens, 2015; Wiens, 2024; 2025). Starting from early minimax formulations (Huber, 1975), this literature develops designs that minimize the worst-case MSE of estimated parameters under departures from an assumed regression model. For instance, Fang & Wiens (2000) propose minimax robust designs for approximately linear models with unknown nonlinear components and heteroscedastic noise, and characterize how optimal designs trade off variance and model misspecification bias.

These robust design methods are fundamentally *static* and inapplicable to online or sequential assignments. Moreover, they target covariate design rather than A/B testing or treatment effect estimation.

## 3. Robust Sequential Experimental Design in Contextual Bandits

This section presents the proposed design in a contextual bandit setting without carryover effects. To illustrate our main idea, we begin with a setting that assumes additive treatment effects. We first derive a bias-variance decomposition of the ATE estimator's MSE to illustrate the effect of model misspecification (Section 3.1). We next apply an orthogonalization technique to obtain an upper bound for the MSE of the ATE estimator (Section 3.2) and introduce a dynamic programming algorithm to optimize this bound (Section 3.3). Finally, we extend our proposal to accommodate interactive treatment effects (Section 3.4).

### 3.1. Bias-Variance Decomposition

Consider a sequential experiment in which $N$ experimental units arrive over time. Each unit $i$ is characterized by a covariate vector $X_i \in \mathbb{R}^p$, whose first component corresponds to the intercept. Without loss of generality, we assume its remaining components are mean zero, as they can always be centered otherwise. For each arriving unit, the policy maker must assign a treatment $A_i \in \{\pm 1\}$, where $A_i = 1$ denotes the new policy and $A_i = -1$ denotes the standard control. Next, an outcome $Y_i$ is observed. In the contextual bandit setting, $\{X_i\}_i$ are independent and identically distributed (i.i.d.). Additionally, $Y_i$ depends solely on the current covariate $X_i$ and assignment $A_i$, being independent of all historical covariates, assignments, and outcomes. Thus, there are no carryover effects in this setting.

We adopt a working linear model, $A\gamma + X^\top \beta$, to approximate the true outcome regression function $m(A, X) = \mathbb{E}(Y|A, X)$. Let $\theta = (\gamma, \beta^\top)^\top \in \mathbb{R}^{p+1}$ denote the model parameter such that the linear model provides the best linear approximation to the regression function,

$$\theta = \arg \min_{\alpha \in \mathbb{R}^{p+1}} \sum_{a \in \{-1, 1\}} \mathbb{E} \left[ m(a, X) - (a, X^\top)\alpha \right]^2. \quad (1)$$

Here, the expectation is taken with respect to the distribution

of $X$, while the action is averaged uniformly over $\{-1, 1\}$. Importantly, we do not assume that the true regression function is linear. Consequently, the approximation error in (1) may be non-negligible. Under this misspecified setting, $\theta$ remains well-defined as the "least-false" parameter in the sense of White (1982).

To explicitly account for the model misspecification, we denote the approximation error by $f(X)$. The data generating process can then be expressed as:

$$Y_i = \gamma A_i + X_i^\top \beta + f(X_i) + e_i, \qquad (2)$$

where $\{e_i\}_i$ are i.i.d. random errors with mean zero and variance $\sigma^2$. We assume each $e_i$ is independent of both $A_i$ and $X_i$.

To illustrate our main idea, we focus on this basic setting in this section. We provide additional discussions on more general settings, including treatment-dependent misspecification $f(X, A)$ and heteroskedastic errors in Appendix D. Our target estimand is the ATE, defined as

$$\text{ATE} = \mathbb{E}[m(1, X) - m(-1, X)],$$

which equals $2\gamma$ under Model (2) where the treatment effect is additive[1]. Consequently, estimating the ATE reduces to estimating the treatment effect coefficient $\gamma$.

We use ordinary least squares (OLS) to estimate $\gamma$. Let $\mathbf{X} \in \mathbb{R}^{N \times p}$ be the design matrix with the $i$th row being $X_i^\top$ and $\mathbf{a} = (A_1, \ldots, A_N)^\top \in \{\pm 1\}^N$ be the treatment allocation vector. A standard calculation in Appendix A.1 shows that the OLS estimator $\widehat{\gamma}_{\mathbf{a}}$ admits the following MSE decomposition,

$$\text{MSE}(\widehat{\gamma}_{\mathbf{a}}) = \frac{\sigma^2}{\mathbf{a}^\top \mathbf{P}_{\mathbf{X}^\perp} \mathbf{a}} + \left( \frac{\mathbf{a}^\top \mathbf{P}_{\mathbf{X}^\perp} f(\mathbf{X})}{\mathbf{a}^\top \mathbf{P}_{\mathbf{X}^\perp} \mathbf{a}} \right)^2, \quad (3)$$

where $f(\mathbf{X}) = (f(X_1), \ldots, f(X_N))^\top$ denotes the vector of approximation errors, and $\mathbf{P}_{\mathbf{X}^\perp}$ denotes the projection onto the orthogonal complement of the column space of $\mathbf{X}$.

The first term in (3) corresponds to the variance of $\widehat{\gamma}_{\mathbf{a}}$, whereas the second term is the squared bias arising from model misspecification. It is natural to consider minimizing (3) to identify the optimal treatment allocation vector $\mathbf{a}$. However, two challenges arise: (i) The approximation error function $f$ is generally unknown. (ii) Minimizing (3) requires to observe the covariates for all units. In our sequential setting, we must determine $A_i$ before future units arrive. We address these challenges in Sections 3.2 and 3.3 using orthogonalization and dynamic programming, respectively.

To conclude this section, we remark that the MSE in (3) is conditional on the observed covariates. This conditioning is

[1]We will relax such an additivity assumption in Section 3.4.

crucial: it allows the treatment assignment that minimizes (3) to adaptively depend on the covariate history to mitigate finite-sample randomness. In contrast, designs that minimize a population-level MSE (e.g., Neyman allocation) would result in rules that depend only on the current covariate and do not explicitly account for finite-sample randomness.

### 3.2. Robust MSE Upper Bounds via Orthogonalization

To eliminate the dependence of the MSE on $f$ in (3), we employ orthogonalization to derive an upper bound that is independent of the specific form of the nonlinear covariate effect.

We begin by approximating $f$ using a sieve representation

$$f(X) \approx \Psi^\top(X)\mathbf{b}, \qquad (4)$$

for a set of basis functions $\Psi(X) = (\psi_1(X), \ldots, \psi_L(X))^\top$ and a coefficient vector $\mathbf{b} \in \mathbb{R}^L$ with the normalization constraint $\|\mathbf{b}\|_2 \leq \eta$ for some $\eta > 0$. It is worth mentioning that we do not estimate $\mathbf{b}$ directly to evaluate the MSE in (3), since its estimation would be prone to large errors when $\Psi$ is high-dimensional. Instead, we seek a robust MSE upper bound that holds uniformly over the space of coefficients.

Our key observation from (3) is that the approximation error $f$ must satisfy the orthogonality condition $N^{-1}\mathbb{E}[\mathbf{X}^\top f(\mathbf{X})] = 0$. This condition follows directly from the definition of $\theta$ as the best linear approximation to $m(a, X)$ (see Equation (1)). Invoking the sieve approximation in (4), we obtain that

$$\mathbb{E}\left[ \frac{1}{N}\mathbf{X}^\top \Psi(\mathbf{X}) \right] \mathbf{b} \approx 0, \qquad (5)$$

where $\Psi(\mathbf{X}) \in \mathbb{R}^{N \times L}$ denotes the matrix of basis functions whose $i$th row is given by $\Psi^\top(X_i)$.

The orthogonality condition in (5) essentially restricts $\mathbf{b}$ to lie in the null space of the matrix $\mathbb{E}[N^{-1}\mathbf{X}^\top \Psi(\mathbf{X})]$. Let $\mathbf{U}$ denote an orthonormal basis for this null space. This enables us to reparameterize the approximation error function $f$ as:

$$f(\mathbf{X}) \approx \eta\, \Psi(\mathbf{X})\mathbf{U}\kappa, \qquad (6)$$

for some vector $\kappa$ such that $\|\kappa\| \leq 1$.

Finally, plugging (6) into the MSE decomposition in (3) leads to the following worst-case upper bound:

$$\text{MSE}(\widehat{\gamma}_{\mathbf{a}}) \leq \frac{\sigma^2}{\mathbf{a}^\top \mathbf{P}_{\mathbf{X}^\perp} \mathbf{a}} + \eta^2 \frac{\|\mathbf{a}^\top \mathbf{P}_{\mathbf{X}^\perp} \Psi(\mathbf{X})\mathbf{U}\|_2^2}{(\mathbf{a}^\top \mathbf{P}_{\mathbf{X}^\perp} \mathbf{a})^2}. \quad (7)$$

We use the upper bound in (7) as the objective function for optimization. Note that the second term depends on $\eta$ – which quantifies the magnitude of the approximation error – and the basis functions $\Psi$, but it remains independent of the unknown coefficient vector $\mathbf{b}$. We relegate the detailed technical derivations to Appendix A.1.

## 3.3. Sequential Design via Dynamic Programming

In our sequential setting, each treatment assignment must be determined before future covariates are observed. To address this, we employ DP to optimize (7).

To simplify the calculation, we observe that the MSE upper bound in (7) depends on the treatment assignments and covariates only through a few summary statistics. More specifically, we define two imbalance statistics,

$$\Delta_i = \sum_{t=1}^{i} a_t X_t \text{ and } \Gamma_i = \sum_{t=1}^{i} a_t \Psi(X_t), \qquad (8)$$

where $\Delta_i$ measures the observed covariate imbalance between the treatment and control groups, and $\Gamma_i$ quantifies the imbalance with respect to the sieve basis functions $\Psi$.

With some calculations detailed in Appendix A.1, we can approximate the MSE upper bound in (7) using the defined imbalance statistics as follows:

$$\underbrace{\frac{\sigma^2}{N - N^{-1}\Delta_N^\top \Sigma^{-1}\Delta_N}}_{\text{variance}} + \underbrace{\frac{\eta^2 \left\| \mathbf{U}^\top (\Gamma_N - \Xi\Sigma^{-1}\Delta_N) \right\|_2^2}{\left( N - N^{-1}\Delta_N^\top \Sigma^{-1}\Delta_N \right)^2}}_{\text{worst-case bias}},$$

$$(9)$$

where $\Sigma$ and $\Xi$ denote certain population-level matrices defined in Proposition A.1 in Appendix A.1.

Crucially, (9) depends on the treatment assignments and covariates solely through the summary statistics $\Delta_N$ and $\Gamma_N$. This property enables us to formulate the sequential allocation as a DP, with $(\Delta_i, \Gamma_i)$ being the Markov state for $i = 1, \ldots, N$. We summarize the result in the following theorem.

**Theorem 1** (Bellman recursion). *Suppose that Assumptions A.1–A.3 hold. Let $Q_N(\Delta_N, \Gamma_N)$ denote the MSE upper bound in (9), which serves as our ultimate objective function. Then the optimal sequential design problem can be formulated as a dynamic program with Markov state $(\Delta_i, \Gamma_i)$ and value functions $\{Q_i\}_{i=1}^{N-1}$ defined through the following Bellman equation. At stage $i$, the value function $Q_i(\Delta_i, \Gamma_i)$ is defined as*

$$\mathbb{E}\Big[ \min_{a \in \{\pm 1\}} Q_{i+1}\big( \Delta_i + a X_{i+1}, \Gamma_i + a\Psi(X_{i+1}) \big) \Big], \quad (10)$$

*for $i = N-1, \cdots, 1$. Consequently, an optimal sequential design can be obtained as follows. At each stage $i$, after observing $X_i$ and given $(\Delta_{i-1}, \Gamma_{i-1})$, choose*

$$a_i^* \in \arg\min_{a \in \{\pm 1\}} Q_i(\Delta_{i-1} + a X_i, \, \Gamma_{i-1} + a\Psi(X_i)) .$$

The DP formulation is conceptually elegant. However, when the dimension of $X$ or $\Psi(X)$ is moderately large, exact

---

**Algorithm 1** Training value function $Q_n$ via Synthetic Rollouts

1: **Input:** A covariate distribution $\mathcal{F}_X$ (or its empirical distribution derived from historical data), a behavior policy $\pi_b$, number of rollouts $B$, value function $Q_{n+1}$.
2: Sample $\{X_{n+1}^{(m)}\}_{m=1}^{M}$ from $\mathcal{F}_X$
3: **for** $b = 1, \ldots, B$ **do**
4:     Sample $\{X_i^{(b)}\}_{i=1}^{n}$ from $\mathcal{F}_X$
5:     Sample $\{a_i^{(b)}\}_{i=1}^{n} \in \{\pm 1\}^n$ from $\pi_b$
6:     Calculate $\Delta_n^{(b)}, \Gamma_n^{(b)}$ by (8)
7:     Calculate the target $J_n^{(b)}$ using Monte Carlo

$$\min_{a \in \{\pm 1\}} \frac{\sum_{m=1}^{M} Q_{n+1}(\Delta_n^{(b)} + a X_{n+1}^{(m)}, \Gamma_n^{(b)} + a\Psi(X_{n+1}^{(m)}))}{M}$$

8: **end for**
9: Train a DNN to compute $Q_n$ using $\{(\Delta_n^{(b)}, \Gamma_n^{(b)})\}_{b=1}^{B}$ as the predictors and $\{J_n^{(b)}\}_{b=1}^{B}$ as the targets.
10: **Output:** $Q_n$.

---

computation of the value functions via backward induction becomes infeasible. To address this, we approximate these value functions using deep neural networks; see Algorithm 1 for details on how these networks are trained using synthetic rollouts. Meanwhile, since the matrices $\Sigma$, $\Xi$, and $\mathbf{U}$ depend solely on the covariate distribution, they can be estimated offline prior to the experiment by leveraging the rich historical data. We summarize our procedure in Algorithm 2, with implementation details provided in Appendix B.1.

---

**Algorithm 2** Robust Sequential Design

1: **Input:** Covariate distribution $\mathcal{F}_X$ (or its empirical distribution from historical data); terminal value function $Q_N$ of (9); Initialize $\Delta_0 = 0, \Gamma_0 = 0$.
2: **for** $n = N-1, \ldots, 1$ **do**
3:     Training value function $Q_n$ by Algorithm 1.
4: **end for**
5: **for** $n = 1, \ldots, N$ **do**
6:     Observe $X_n$ from $\mathcal{F}_X$.
7:     Maintain the historical summaries $(\Delta_{n-1}, \Gamma_{n-1})$ from past assignments.
8:     Calculate the optimal treatment $a_n^*$ by

$$\arg\min_{a \in \{-1, 1\}} Q_n\Big( \Delta_{n-1} + a X_n, \Gamma_{n-1} + a\Psi(X_n) \Big).$$

9:     Update $\Delta_n = \Delta_{n-1} + a_n^* X_n$ and $\Gamma_n = \Gamma_{n-1} + a_n^* \Psi(X_n)$ according to Eq (8).
10: **end for**
11: **Output:** Optimal treatments $\{a_n^*\}_{n=1}^{N}$.

---

## 3.4. Robust Design with Interactive Treatment Effects

In this section, we extend the proposed design to settings with treatment-covariate interactions. To accommodate treatment effects that vary across experimental units, we adopt the following working linear model:

$$\mathbb{E}(Y|A, X) \approx AX^\top \xi + X^\top \beta. \qquad (11)$$

Compared to the additive model in (2), the treatment effect in (11) interacts with the covariates and becomes heterogeneous across experimental units.

Assuming that the underlying model $\mathbb{E}(Y|A, X)$ takes the form $AX^\top \xi + X^\top \beta + f(X)$ for some approximation error $f(X)$, it can be shown that the ATE equals $2\mathbb{E}(X^\top \xi)$. Consequently, the parameter $\xi$ can be estimated via OLS under the specified working model to construct a plug-in estimator for the ATE. To handle model misspecification, we apply an orthogonalization procedure analogous to that in Section 3.2 to derive an upper bound on the MSE of the resulting ATE estimator. We further employ a DP algorithm, similar to the one in Section 3.3, for online allocation. To maintain conciseness, the explicit MSE upper bound and the associated Bellman recursion are relegated to Proposition A.2 and Theorem A.1 in Appendix A.2.

# 4. Extensions to Dynamic Settings

This section generalizes our proposal developed in Section 3 to dynamic settings. In many practical applications, treatments are repeatedly assigned throughout a given day, resulting in temporally dependent data where prior assignments exert carryover effects on future outcomes.

To formalize this problem, we consider an experiment conducted over $N$ days, with each day partitioned into $T$ non-overlapping time intervals. At the beginning of the $t$th interval on day $i$, the policy maker observes a covariate vector $X_{it} \in \mathbb{R}^p$ and assigns a binary treatment $A_{it} \in \{\pm 1\}$. The corresponding outcome, $Y_{it}$, is then observed at the end of the interval.

We assume that data are independent across days and that the outcome depends exclusively on the current treatment-covariate pair – an assumption commonly imposed in RL (Sutton & Barto, 2018). While the outcome is conditionally independent of the history, the evolution of the covariates depends on the past treatments on that day. This differs from the contextual bandit setting in Section 3, enabling us to capture complex carryover effects while maintaining a simple outcome regression model for treatment effect estimation.

To illustrate our main idea, we begin by assuming a linear outcome model with time-varying coefficients and interac-tive treatment effects (Luo et al., 2024):

$$\mathbb{E}(Y_{it} \mid A_{it}, X_{it}) = A_{it} X_{it}^\top \xi_t + X_{it}^\top \beta_t, \qquad (12)$$

where $\theta_t = (\xi_t^\top, \beta_t^\top)^\top \in \mathbb{R}^{2p}$ denotes the vector of time-varying coefficients. In this dynamic setting, our target ATE is defined as

$$\text{ATE} := \frac{1}{T} \Big[ \mathbb{E}^{(1)}\Big( \sum_{t=1}^{T} Y_t \Big) - \mathbb{E}^{(-1)}\Big( \sum_{t=1}^{T} Y_t \Big) \Big],$$

where $\mathbb{E}^{(1)}$ (resp. $\mathbb{E}^{(-1)}$) denotes the expectation under the counterfactual world in which action $+1$ (resp. $-1$) is applied at all times.

Under the linear model assumption in (12), we estimate $\{\theta_t\}_t$ via OLS and construct a plug-in estimator for the ATE. Unlike the contextual bandit setting, however, the MSE of this estimator is substantially more complex, since its variance arises not only from estimating the regression coefficients, but also from learning how covariates evolve as a function of past actions. The latter source of variability is extremely difficult to characterize analytically.

To obtain a tractable objective, we approximate the MSE using a surrogate that ignores the latter source of variation and focuses solely on the estimation error induced by the OLS estimators. We further allow the outcome model (12) to be misspecified and apply the orthogonalization procedure to upper bound this surrogate MSE, leading to the following objective function,

$$\frac{1}{T} \sum_{t=1}^{T} \Big[ \frac{\mathbf{u}_t^\top G_t^{-1} \mathbf{u}_t}{T} + \nu_t^2 \big\| \mathbf{U}_t^\top H_t G_t^{-1} \mathbf{u}_t \big\|_2^2 \Big], \qquad (13)$$

where $\mathbf{u}_t = (\mathbb{E}^{(1)} X_t^\top + \mathbb{E}^{(-1)} X_t^\top, \mathbb{E}^{(1)} X_t^\top - \mathbb{E}^{(-1)} X_t^\top)^\top$ characterizes how covariates evolve over time, matrices $G_t$ and $H_t$ depend on the treatment–covariate history and enable the optimization of (13) to incorporate all past observations, $\nu_t^2 = \frac{\eta_t^2}{\sigma_t^2} > 0$ is a time-specific tuning parameter and $\mathbf{U}_t$ denotes the orthonormal basis, defined as in Section 3. To save space, we defer all technical derivations to Appendix A.3.

Similar to Section 3, we can apply DP to minimize (13) to derive an optimal sequential design. However, DP in dynamic settings is computationally infeasible: there are $N \times T$ decision stages in total, and backward induction must be performed over all $N \times T$ stages. For instance, in ride-sharing applications, experiments often last for two weeks, with each day divided into 30-minute or 1-hour intervals, resulting in approximately $300 - 600$ decision stages (Xu et al., 2018; Shi et al., 2023b).

To address this challenge, we propose a hierarchical design that explicitly exploits the inherent structure of the experiment. At a high level, our global optimization problem is decomposed into two subproblems:

**(i)** A macro-level objective that applies DP to perform backward induction over the $N$-day horizon, yielding a day-level objective function for each day.

**(ii)** A micro-level objective that leverages RL to determine optimal treatment decisions across the $T$ intervals within each day by maximizing the negative day-level loss, equivalently minimizing the day-level objective.

We elaborate on the two sub-problems below.

**Across-day DP**. We treat each day as a single decision stage with a joint action space of assigning $T$ treatments within that day. Because covariates and outcomes are independent across days, this macro-level problem reduces to the bandit problem studied in Section 3, with $N$ stages. As a result, DP can be efficiently applied to compute the day-level objective functions via backward induction.

More specifically, similar to (8), we introduce the following across-day summary statistics for each time $t = 1, \ldots, T$:

$$B_{it} := \sum_{j=1}^{i} X_{jt} X_{jt}^\top, \quad C_{it} := \sum_{j=1}^{i} A_{jt} X_{jt} X_{jt}^\top,$$

$$D_{it} := \sum_{j=1}^{i} \Psi(X_{jt}) X_{jt}^\top, \quad F_{it} := \sum_{j=1}^{i} A_{jt} \Psi(X_{jt}) X_{jt}^\top.$$

These statistics yield our day-level Markov state

$$\mathcal{S}_i := \{(B_{it}, C_{it}, D_{it}, F_{it})\}_{t=1}^{T},$$

which serves as the sufficient information for the day-level optimal policy. Let $\pi \in \Pi$ denote a day-level policy within the policy class $\Pi$ that determines the treatment sequence for an entire day, the following theorem formally proves this result.

**Theorem 2** (Across-day Bellman recursion). *Let $\mathcal{Q}_N(\mathcal{S}_N)$ denote the MSE upper bound in (13) which serves as our ultimate objective function. The optimal design problem*

$$\inf_{\pi_1, \ldots, \pi_N \in \Pi} \mathbb{E}\big[\mathcal{Q}_N(\mathcal{S}_N)\big]$$

*admits a dynamic programming formulation with Markov state $\mathcal{S}_i$. Specifically, define value functions $\{\mathcal{Q}_i\}_{i=0}^{N-1}$ by*

$$\mathcal{Q}_{i-1}(\mathcal{S}_{i-1}) = \mathbb{E}\Big[\inf_{\pi \in \Pi} \mathcal{Q}_i\big(\mathcal{S}_i\big) | \mathcal{S}_{i-1}\Big], \quad (14)$$

*for $i = N, \ldots, 1$. Moreover, each optimal policy $\pi_i^*$ can be computed by solving the above optimization.*

Theorem 2 motivates a backward induction procedure to compute the value functions $\{\mathcal{Q}_i\}_i$ for $i = N, N-1, \cdots, 1$, which serve as our day-level objectives. However, an exact computation of these functions is computationally intractable due to the high-dimensional state space. Consequently, we employ Monte Carlo rollouts to approximate the Q function based on (13). Refer to Algorithms 4 and 5 in Appendix B.3 for details.

**Within-day RL**. Given an approximated value function $\mathcal{Q}_i$ computed via backward induction, solving the right-hand side of (14) can be readily formulated as a finite-horizon MDP. This motivates our application of RL to efficiently learn the optimal policy $\pi_i^*$. To be more specific, we define the state-action-reward triplets in this MDP below and prove this assertion in Theorem 3.

- *State*: The state at each time $t$ is set to $\mathcal{F}_t = \{\mathcal{S}_{i-1}, \mathcal{H}_{i,t}\}$, where $\mathcal{S}_{i-1}$ corresponds to the Markov state at the $(i-1)$th day and $\mathcal{H}_{i,t}$ corresponds to the covariate-treatment pairs collected up to time $t$.

- *Action*: An action $A_{it} \in \{\pm 1\}$ is then generated according to a within-day policy $A_{it} \sim \pi_{i,t}(\cdot \mid \mathcal{F}_t)$.

- *Reward*: The day-level objective $\mathcal{Q}_i(\mathcal{S}_i)$ can be approximated via Monte Carlo rollouts. Owing to the additive structure of (13), these approximations decompose into per-step rewards $R_t$ (see Appendix B.3 for details).

**Theorem 3** (Within-day Bellman recursion). *The right-hand side of (14) forms a finite-horizon MDP with state-action-reward triplets defined above.*

As a result, the optimal within-day policy $\pi_i^* = \{\pi_{i,t}^*\}_t$ can be computed via RL. In our implementation, we adopt an actor-critic algorithm and relegate all implementation details to Appendix B.3.

Our global objective function (13) is minimized through a backward procedure over days: once $\pi_i^*$ is determined, we approximate $Q_{i-1}$ using Monte Carlo rollouts and optimize the RL sub-problem for day $i-1$ to compute $\pi_{i-1}^*$. This recursive process continues until the complete sequence of optimal policies $\{\pi_i^*\}_{i=1}^N$ has been obtained.

## 5. Numerical Experiments

We evaluate the finite-sample performance of our method through a series of numerical experiments. We use synthetic data from both contextual bandit (Section 5.1) and dynamic (Section 5.2) settings, along with a real-world dataset (Section 5.3). Our code is available at RSD-for-AB-Testing.

### 5.1. Contextual Bandits

**DGP.** We consider two data generating processes (DGPs): one with additive treatment effects and the other with interactive treatment effects. Both DGPs are nonlinear so that the linear working model is misspecified. Covariates are drawn from a zero-mean bivariate normal distribution, and the experiment duration is set to $N \in \{21, 28, 35, 42\}$ days.

For further details regarding the experimental setup, refer to Appendix C.1.

**Baseline algorithms.** We compare our proposed robust sequential design (denoted by *RSD*) against several baseline approaches: (i) A non-robust variant (*NRD*) that optimizes only the variance component of the MSE objective. Under additive treatment effects, this variant reduces to the proposal by Bhat et al. (2020). (ii) A random design (*RND*) that assigns treatment independently at random each day; (iii) A switchback design (*SBD*), which alternates treatment assignments (e.g., $+1, -1, +1, -1, \ldots$, with a random starting sign); (iv) A Neyman balanced design (*NBD*), implemented via a biased-coin procedure that sequentially targets the Neyman-optimal allocation. In our homoskedastic setting, this targets a balanced allocation (Cai & Rafi, 2024). (v) A Bayesian biased-coin design (*BBD*), which adaptively adjusts assignment probabilities based on accumulated imbalance (Ball et al., 1993).

**Results.** The upper panels of Figure 2 (top row) present the average empirical MSE of the ATE estimators under all designs, aggregated over 400 simulation replications under both additive and interactive treatment effects. We make the following observations: (i) The proposed **RSD** consistently achieves the lowest MSE, which decays as the experiment duration $N$ increases, confirming its large-sample consistency. Furthermore, its advantage is more pronounced in small-sample settings, highlighting its finite-sample-awareness. (ii) **NRD** does not consistently perform well. While it accounts for finite-sample randomness and utilizes DP for long-term optimization, it fails to guard against model misspecification. (iii) Finally, **RND**, **SBD**, **NBD**, and **BBD** suffer from significantly large MSEs, as they are either not finite-sample aware or rely on sequential but myopic strategies that do not explicitly account for future outcomes.

### 5.2. Dynamic Settings

**DGP.** We consider a dynamic setting where each day is modeled as an MDP episode with $T \in \{6, 12\}$ time intervals. At each time $t$, the time-varying covariates are Gaussian, while the response follows a nonlinear model to induce misspecification. We vary the sample size $N \in \{21, 28, 35, 42\}$ and consider three levels of nonlinearity in the response model, corresponding to three degrees of model misspecification. For details of the experimental setup, see Appendix C.2.

**Baseline algorithms.** We compare the proposed *RSD* against: (i) a non-robust variant *NRD* adapted to the dynamic setting; (ii) *SBD* that is widely used in practice (Bojinov et al., 2023); (iii) *TMDP* and *NMDP* proposed by Li et al. (2023) that extend Neyman allocation to the dynamic setting; (iv) *RND*.

**Results.** The lower panels of Figure 2 (bottom row) and

Figure 4 in Appendix C.2 present results across different time horizons and degrees of nonlinearity, averaged over 1000 simulation replications. Overall, the proposed design remains the most competitive across all settings.

### 5.3. Real-Data-Based Simulation

**DGP.** We use data from a leading ride-sharing company to evaluate different experimental designs. The covariates include (i) the daily number of order requests and (ii) drivers' total online time at the beginning of each day, which proxy market demand and supply. The treatment corresponds to order-dispatch policies, and the outcome is the average daily driver income. Since the underlying data-generating process is unknown, we follow standard practice (e.g., Wen et al. (2025)) by constructing a simulation environment calibrated to the real data and using it for evaluation. Details are provided in Appendix C.3.

**Results.** We compare the proposed method with the same baselines as in Section 5.1. Figure 3 summarizes results over 400 simulation replications. The trends closely mirror those in Section 5.1: our design consistently outperforms all baselines. Since this evaluation is grounded in real-world data, the gains further underscore the practical effectiveness of our approach.

## 6. Conclusion

We propose a robust sequential experimental design framework for A/B testing that covers both contextual bandits and dynamic settings with carryover effects. Our approach (i) achieves robustness to model misspecification via orthogonalization, reducing design to minimizing a tractable upper bound on the MSE of ATE; (ii) mitigates finite-sample covariate imbalance while optimizing long-horizon objectives via dynamic programming; and (iii) handles carryover dynamics through a nested scheme that combines across-day DP with within-day RL.

## Acknowledgements

We thank the anonymous referees and the meta-reviewer for their constructive comments, which have significantly improved this manuscript. Qianglin Wen's research was supported by the National Key R&D Program of China (Grant No. 2022YFA1003701). Xiangkun Wu's research was supported by the National Key Research and Development Program of China (Grant No. 2024YFC2511003). Yingying Zhang's research was supported by the National Natural Science Foundation of China (Grant No. 12471280). Ting Li's research was supported by the National Natural Science Foundation of China (Grant No. 12571304), the Shanghai Pujiang Program (Grant No. 24PJIC030), the CCF–DiDi GAIA Collaborative Research Funds, and the

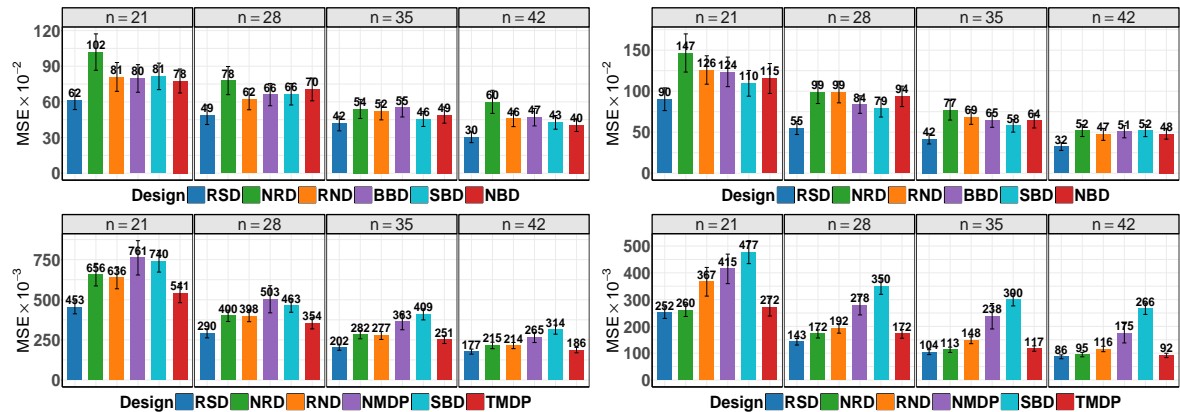

*Figure 2.* Empirical MSE (95% CI): under the contextual bandits with additive treatment effects (top left), with interactive treatment effects (top right); under the dynamic settings with large bias: with $T = 6$ (bottom left), with $T = 12$ (bottom right).

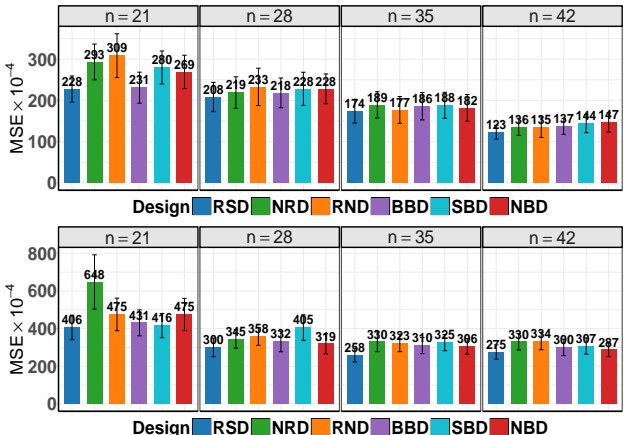

*Figure 3.* Empirical MSE (95% CI) based on the real-data-based simulation: with additive treatment effects (top), with interactive treatment effects (bottom).

Program for Innovative Research Team of Shanghai University of Finance and Economics.

## Impact Statement

This paper presents a robust sequential experimental design algorithm for A/B testing. Compared with standard randomization and other treatment allocation methods, our approach improves the reliability and accuracy of treatment-effect estimation under model misspecification. As a methodological contribution intended for settings where A/B testing is already routine, we do not expect the proposed approach itself to introduce additional negative societal impacts beyond those associated with A/B testing in general. Nevertheless, A/B testing can affect users, workers, and market participants when deployed in real-world platforms, and practitioners should ensure that experiments comply with applicable privacy, fairness, transparency, and safety requirements.

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

# A. Technical Details

In this section, we provide the technical details for the results in Sections 3 and 4. Specifically, Sections A.1 and A.2 contain the derivations and proofs for the robust design in the contextual bandits without and with treatment-covariate interactions, respectively, while Section A.3 presents the technical details for the hierarchical design in the dynamic setting.

## A.1. Robust Design in Contextual Bandits without Interactions

In this subsection, we first derive the explicit conditional MSE and its upper bound, and then show that the resulting surrogate objective admits a dynamic programming solution. We begin by summarizing the required assumptions.

**Assumption A.1.** (Noise independence) The noise variables $\{e_i\}_{i=1}^N$ are i.i.d. with $\mathbb{E}e_i = 0$ and $\mathrm{Var}(e_i) = \sigma^2$.

**Assumption A.2.** (Non-singular covariance matrix) The covariance matrix $\mathbb{E}(\mathbf{X}\mathbf{X}^\top)$ is positive definite.

**Assumption A.3.** (Hölder smoothness & bounded sieves) The nonlinear component $f(\cdot)$ belongs to a Hölder class $\Lambda(d, c)$ (defined at the end of this subsection), which admits a uniformly bounded sieve approximation. Moreover, $f(\cdot)$ is uniformly bounded in the sense that $\sup_x |f(x)| \leq \eta_f^2$ for some constant $\eta_f > 0$.

First, Assumption A.1 is mild and standard in experimental design for linear models (Bhat et al., 2020; López-Fidalgo, 2023). Second, Assumption A.2 rules out perfect multicollinearity and ensures that the linear coefficients are identifiable from the data. Finally, we assume that the nonlinear component $f(\cdot) \in \Lambda(d, c)$, which is a common smoothness condition in nonparametric regression (Györfi et al., 2002; Chen, 2007). Moreover, we control the magnitude of the nonlinear component and restrict the sieve space to be uniformly bounded. These conditions imply that the sieve coefficients $\mathbf{b}$ are bounded. As a result, we can obtain an explicit upper bound on the MSE (Wiens, 2000; 2009; Kong & Wiens, 2015).

**Derivation of conditional MSE.** For a fixed treatment assignment policy $\mathbf{a} \in \{\pm 1\}^N$, the OLS estimator admits the closed-form expression

$$\widehat{\theta}_\mathbf{a} = (\mathbf{Z}^\top \mathbf{Z})^{-1} \mathbf{Z}^\top \mathbf{Y},$$

where $\mathbf{Z} = [\mathbf{a}, \mathbf{X}] \in \mathbb{R}^{N \times (p+1)}$ and $\mathbf{Y} = (Y_1, \ldots, Y_N)^\top \in \mathbb{R}^N$. Let $\mathbf{u} = (1, 0, \ldots, 0)^\top \in \mathbb{R}^{p+1}$ denote the selector vector corresponding to the treatment effect, so that $\widehat{\gamma}_\mathbf{a} = \mathbf{u}^\top \widehat{\theta}_\mathbf{a}$. Conditioning on $(\mathbf{a}, \mathbf{X})$, the MSE of $\widehat{\gamma}_\mathbf{a}$ can be written as

$$
\begin{aligned}
\mathrm{MSE}(\widehat{\gamma}_\mathbf{a}) &= \mathbf{u}^\top \left[ (\mathbf{Z}^\top \mathbf{Z})^{-1} \mathbf{Z}^\top \mathbb{E}(\mathbf{e}\mathbf{e}^\top) \mathbf{Z} (\mathbf{Z}^\top \mathbf{Z})^{-1} + (\mathbf{Z}^\top \mathbf{Z})^{-1} \mathbf{Z}^\top (\boldsymbol{f}\boldsymbol{f}^\top) \mathbf{Z} (\mathbf{Z}^\top \mathbf{Z})^{-1} \right] \mathbf{u} \\
&= \underbrace{\sigma^2 \, \mathbf{u}^\top (\mathbf{Z}^\top \mathbf{Z})^{-1} \mathbf{u}}_{I_1} + \underbrace{\mathbf{u}^\top (\mathbf{Z}^\top \mathbf{Z})^{-1} \mathbf{Z}^\top (\boldsymbol{f}\boldsymbol{f}^\top) \mathbf{Z} (\mathbf{Z}^\top \mathbf{Z})^{-1} \mathbf{u}}_{I_2},
\end{aligned}
\tag{A.1}
$$

where $\boldsymbol{f} := f(\mathbf{X}) = (f(X_1), \ldots, f(X_N))^\top \in \mathbb{R}^N$. To further analyze these terms, note that

$$\mathbf{Z}^\top \mathbf{Z} = \begin{pmatrix} \mathbf{a}^\top \mathbf{a} & \mathbf{a}^\top \mathbf{X} \\ \mathbf{X}^\top \mathbf{a} & \mathbf{X}^\top \mathbf{X} \end{pmatrix} = \begin{pmatrix} N & \Delta_N^\top \\ \Delta_N & \mathbf{X}^\top \mathbf{X} \end{pmatrix},$$

where $\Delta_N := \mathbf{X}^\top \mathbf{a} = \sum_{t=1}^N a_t X_t \in \mathbb{R}^p$, which quantifies the imbalance in treatment assignments and covariates, as illustrated in (8). By the block matrix inversion formula, we obtain

$$(\mathbf{Z}^\top \mathbf{Z})^{-1} = \begin{pmatrix} \alpha & -\alpha \Delta_N^\top (\mathbf{X}^\top \mathbf{X})^{-1} \\ -\alpha (\mathbf{X}^\top \mathbf{X})^{-1} \Delta_N & (\mathbf{X}^\top \mathbf{X})^{-1} + \alpha (\mathbf{X}^\top \mathbf{X})^{-1} \Delta_N \Delta_N^\top (\mathbf{X}^\top \mathbf{X})^{-1} \end{pmatrix},$$

where $\alpha = 1/\mathbf{a}^\top \mathbf{P}_{\mathbf{X}^\perp} \mathbf{a}$ and $\mathbf{P}_{\mathbf{X}^\perp} = \mathbf{I} - \mathbf{X}(\mathbf{X}^\top \mathbf{X})^{-1} \mathbf{X}^\top$. Substituting this expression into (A.1) yields (3).

**Derivation of an upper bound for the MSE.** We first derive an explicit upper bound for the conditional MSE in (3). To facilitate optimization, we then construct a surrogate version of this upper bound by replacing certain quantities with their population counterparts.

**Proposition A.1.** *Suppose that $\theta$ is defined as the solution to (1), and that Assumptions A.1–A.3 hold. Then, for any fixed treatment assignment policy $\mathbf{a}$, the finite-sample MSE of $\widehat{\gamma}_\mathbf{a}$ admits the following upper bound, for some constant $\eta > 0$:*

$$\mathrm{MSE}(\widehat{\gamma}_\mathbf{a}) \leq \underbrace{\frac{\sigma^2}{N - N^{-1} \Delta_N^\top \Sigma_N^{-1} \Delta_N}}_{variance} + \underbrace{\frac{\eta^2 \left\| \mathbf{U}^\top (\Gamma_N - \Xi_N \Sigma_N^{-1} \Delta_N) \right\|_2^2}{\left( N - N^{-1} \Delta_N^\top \Sigma_N^{-1} \Delta_N \right)^2}}_{worst\text{-}case\ bias}.
\tag{A.2}$$

*Here,*

$$\Sigma_N = \frac{1}{N} \sum_{i=1}^N X_i X_i^\top \in \mathbb{R}^{p \times p}, \qquad \Xi_N = \frac{1}{N} \sum_{i=1}^N \Psi(X_i) X_i^\top \in \mathbb{R}^{L \times p},$$

$\mathbf{U} \in \mathbb{R}^{L \times (L-p)}$ *is a matrix whose columns form an orthonormal basis for the orthogonal complement of the column space of* $\mathbf{C} = \mathbb{E}\left[N^{-1}\mathbf{\Psi}^\top(\mathbf{X})\mathbf{X}\right]$, *and* $\Delta_N$, $\Gamma_N$ *are the imbalance statistics defined in* (8).

*Proof.* Recall the conditional MSE decomposition in (3). We first derive an upper bound for the bias term $I_2$ in (A.1) via a sieve-based representation. Under Assumption A.3, the nonlinear component $f(\cdot)$ admits the approximation

$$f(\mathbf{X}) = \eta \, \mathbf{\Psi}(\mathbf{X}) \mathbf{U} \boldsymbol{\kappa} + r_N, \qquad \|\boldsymbol{\kappa}\|_2 \leq 1,$$

where the approximation error satisfies $\sup_x |r_N| = O(L^{-d/p})$ by Lemma 5 in Stone (1985), and can be dominated by the worst-case bound for sufficiently large $L$ and $\mathbf{U}$ spans the orthogonal complement of the column space of $\mathbf{C} = \mathbb{E}[N^{-1}\mathbf{\Psi}^\top(\mathbf{X})\mathbf{X}]$. In particular, note that $\mathbf{C}$ is of full rank $p$ and let $\mathbf{C} = \tilde{\mathbf{U}}_{L \times p} \mathbf{\Lambda}_{p \times p} \mathbf{V}_{p \times p}^\top$, be the singular value decomposition, with $\tilde{\mathbf{U}}^\top \tilde{\mathbf{U}} = \mathbf{V}^\top \mathbf{V} = \mathbf{I}_p$, and $\mathbf{\Lambda}$ diagonal and invertible. Augment $\tilde{\mathbf{U}}$ by $\mathbf{U}_{L \times (L-p)}$ in such a way that $[\tilde{\mathbf{U}} : \mathbf{U}]_{L \times L}$ is orthogonal. Substituting this representation into (3) yields

$$\left(\frac{\mathbf{a}^\top \mathbf{P}_{\mathbf{X}^\perp} f(\mathbf{X})}{\mathbf{a}^\top \mathbf{P}_{\mathbf{X}^\perp} \mathbf{a}}\right)^2 \leq \sup_{\|\boldsymbol{\kappa}\|_2 \leq 1} \frac{\eta^2 \left\| \mathbf{a}^\top \mathbf{P}_{\mathbf{X}^\perp} \mathbf{\Psi}(\mathbf{X}) \mathbf{U} \boldsymbol{\kappa} \right\|_2^2}{(\mathbf{a}^\top \mathbf{P}_{\mathbf{X}^\perp} \mathbf{a})^2}.$$

Taking the supremum over all admissible $\boldsymbol{\kappa}$ gives

$$\left(\frac{\mathbf{a}^\top \mathbf{P}_{\mathbf{X}^\perp} f(\mathbf{X})}{\mathbf{a}^\top \mathbf{P}_{\mathbf{X}^\perp} \mathbf{a}}\right)^2 \leq \frac{\eta^2 \left\| \mathbf{a}^\top \mathbf{P}_{\mathbf{X}^\perp} \mathbf{\Psi}(\mathbf{X}) \mathbf{U} \right\|_2^2}{(\mathbf{a}^\top \mathbf{P}_{\mathbf{X}^\perp} \mathbf{a})^2}. \tag{A.3}$$

Consequently, by combining the variance term in (3) with the right-hand side of (A.3), we obtain the explicit worst-case upper bound on the MSE in (7). To streamline notation, define

$$\mathbf{O}_1 := \mathbf{a}^\top \mathbf{P}_{\mathbf{X}^\perp} \mathbf{a}, \qquad \mathbf{O}_2 := \left\| \mathbf{a}^\top \mathbf{P}_{\mathbf{X}^\perp} \mathbf{\Psi}(\mathbf{X}) \mathbf{U} \right\|_2^2.$$

Then we can rewrite (7) as

$$\mathrm{MSE}(\hat{\gamma}_\mathbf{a}) \leq \frac{\sigma^2}{\mathbf{O}_1} + \frac{\eta^2 \mathbf{O}_2}{\mathbf{O}_1^2}.$$

A key observation is that both $\mathbf{O}_1$ and $\mathbf{O}_2$ admit closed-form expressions in terms of low-dimensional imbalance summaries. For $\mathbf{O}_1$, a standard projection argument gives

$$\mathbf{O}_1 = N - \frac{1}{N} \Delta_N^\top \Sigma_N^{-1} \Delta_N,$$

where

$$\Delta_N = \sum_{i=1}^N a_i X_i, \quad \Sigma_N = \frac{1}{N} \sum_{i=1}^N X_i X_i^\top.$$

Here, $\Delta_N$ captures both the count imbalance and the covariate imbalance, as in Bhat et al. (2020), and $\Sigma_N$ denotes the second-moment matrix of the covariates.

Similarly, for $\mathbf{O}_2$ we obtain

$$\mathbf{O}_2 = \left\| \mathbf{U}^\top \Gamma_N - \mathbf{U}^\top \Xi_N \Sigma_N^{-1} \Delta_N \right\|_2^2,$$

where

$$\Gamma_N = \mathbf{\Psi}^\top(\mathbf{X})\mathbf{a} = \sum_{i=1}^N a_i \boldsymbol{\psi}(X_i) \in \mathbb{R}^L, \quad \Xi_N = \frac{1}{N} \mathbf{\Psi}^\top(\mathbf{X})\mathbf{X} = \frac{1}{N} \sum_{i=1}^N \boldsymbol{\psi}(X_i) X_i^\top \in \mathbb{R}^{L \times p}.$$

The vector $\Gamma_N$ captures imbalance in the sieve features, which arises naturally from the robustification of the bias term. The proof is hence completed. $\qquad \square$

**Surrogate Objective.** To facilitate optimization, we invoke a law-of-large-numbers approximation and replace these sample quantities by their population counterparts, namely $\Sigma_N \approx \Sigma$, and $\Xi_N \approx \Xi$. This approximation yields the following surrogate upper bound on the MSE:

$$\frac{\sigma^2}{N - N^{-1}\Delta_N^\top \Sigma^{-1}\Delta_N} + \frac{\eta^2 \left\| \mathbf{U}^\top (\Gamma_N - \Xi\Sigma^{-1}\Delta_N) \right\|_2^2}{\left( N - N^{-1}\Delta_N^\top \Sigma^{-1}\Delta_N \right)^2}$$

where $\Sigma = \mathbb{E}[XX^\top]$ and $\Xi = \mathbb{E}[\psi(X)X^\top]$. Relative to Bhat et al. (2020), the imbalance statistic $\Delta_N$ retains its classical roles, while the worst-case bias introduces an additional feature-imbalance term $\Gamma_N$. Then we obtain the approximate upper bound in (9).

**Proof of Theorem 1.** Now the sequential design problem is given by

$$(P1) := \min_{\mathbf{a}} \quad \mathbb{E}\left[ \frac{\sigma^2}{N - N^{-1}\Delta_N^\top \Sigma^{-1}\Delta_N} + \frac{\eta^2 \left\| \mathbf{U}^\top (\Gamma_N - \Xi\Sigma^{-1}\Delta_N) \right\|_2^2}{\left( N - N^{-1}\Delta_N^\top \Sigma^{-1}\Delta_N \right)^2} \right]$$
$$\text{s.t.} \quad a_i \in \{\pm 1\}, \quad a_i \text{ is } \mathcal{F}_i\text{-measurable}, \quad i = 1, \dots, N,$$

where the expectation is taken with respect to the covariate process and $\mathcal{F}_i$-measurable means that at each time $i$, the allocation $a_i$ must be made based only on the first $i$ covariates and any prior allocations.

We next show that the problem (P1) admits a Bellman recursion and can therefore be solved via dynamic programming.

*Proof.* For the problem (P1), we note that for the last step $N$:

$$Q_N(\Delta_N, \Gamma_N) = \frac{\sigma^2}{N - N^{-1}\Delta_N^\top \Sigma^{-1}\Delta_N} + \frac{\eta^2 \left\| \mathbf{U}^\top (\Gamma_N - \Xi\Sigma^{-1}\Delta_N) \right\|_2^2}{\left( N - N^{-1}\Delta_N^\top \Sigma^{-1}\Delta_N \right)^2}.$$

Similar to the proof of Proposition 2 in Bhat et al. (2020), we consider the $N$-step MDP:

- The state at decision stage $i$ is defined as $S_i = (\Delta_{i-1}, \Gamma_{i-1}, X_i)$. The terminal state is $S_{N+1} = (\Delta_N, \Gamma_N)$. The state space is $\mathbb{S}_i \subseteq \mathbb{R}^{2p+L}$ for non-terminal decision stages and $\mathbb{S}_{N+1} \subseteq \mathbb{R}^{p+L}$.

- The set of available actions is $\{-1, 1\}$.

- At state $S_i$ if action $a_i$ is chosen, the state $S_{i+1}$ is given by $(\Delta_{i-1} + a_i X_i, \Gamma_{i-1} + a_i \Psi(X_i), X_{i+1})$. After $N$ actions, the terminal state is $S_{N+1} = (\Delta_N, \Gamma_N)$.

- There is no per step reward and the terminal loss is $\mathbb{S}_{N+1} \to Q_N(\Delta_N, \Gamma_N)$.

Note that the MDP has a finite horizon, and the set of actions available at any decision stage is finite (specifically, of size 2). The problem (P1) is just a terminal loss minimization MDP. It follows from Proposition 4.2.1 in Bertsekas (2022) that there exists a policy $a^*$ that achieves the minimum expected loss. Furthermore, there exists a set of functions $J_i^* : \mathbb{S}_i \longrightarrow \mathbb{R}$ such that $J_i^*(s_i)$ is the loss conditioned on $S_i = s_i$. Trivially,

$$J_{N+1}^*(\Delta_N, \Gamma_N) = Q_N(\Delta_N, \Gamma_N).$$

By the Bellman optimality equation for finite-horizon time-dependent MDPs, we can obtain

$$J_i^*(\Delta_{i-1}, \Gamma_{i-1}, X_i) = \min_{u \in \{-1,1\}} \mathbb{E}[J_{i+1}^*(\Delta_{i-1} + uX_i, \Gamma_{i-1} + u\Psi(X_i), X_{i+1})]. \tag{A.4}$$

Consequently, an optimal policy $a_i^*$ can be chosen to satisfy

$$a_i^* = \arg\min_{u \in \{-1,1\}} \mathbb{E}[J_{i+1}^*(\Delta_{i-1} + uX_i, \Gamma_{i-1} + u\Psi(X_i), X_{i+1})]. \tag{A.5}$$

Let,

$$Q_i(\Delta_i, \Gamma_i) = \mathbb{E}[J_{i+1}^*(\Delta_i, \Gamma_i, X_{i+1})]. \tag{A.6}$$

Combining (A.4) and (A.6),

$$Q_i(\Delta_i, \Gamma_i) = \mathbb{E}[\min_{u \in \{-1,1\}} Q_{i+1}(\Delta_i + uX_{i+1}, \Gamma_i + u\Psi(X_{i+1}))]. \tag{A.7}$$

Furthermore, according to (A.5) and (A.6), we can get

$$a_i^* = \arg\min_{u \in \{-1,1\}} Q_i(\Delta_{i-1} + uX_i, \Gamma_{i-1} + u\Psi(X_i)).$$

This proves the dynamic programming procedure. $\qquad\square$

**Definition of Hölder class.** Let $\psi(\cdot)$ be an arbitrary scalar-valued function on $\mathcal{X} \subseteq \mathbb{R}^p$. For a $p$-tuple $\alpha = (\alpha_1, \ldots, \alpha_p)^\top$ of nonnegative integers, let $D^\alpha$ denote the differential operator:

$$D^\alpha \psi(x) = \frac{\partial^{\|\alpha\|_1} \psi(x)}{\partial x_1^{\alpha_1} \cdots \partial x_p^{\alpha_p}}.$$

Here, $x_j$ denotes the $j$th component of $x$. For any $d > 0$, let $[d]$ denote the largest integer strictly smaller than $d$. Define the class of $d$-smooth functions as follows:

$$\Lambda(d, c) = \left\{ \psi : \sup_{\|\alpha\|_1 \le [d]} \sup_{x \in \mathcal{X}} |D^\alpha \psi(x)| \le c, \quad \sup_{\|\alpha\|_1 = [d]} \sup_{\substack{x,y \in \mathcal{X} \\ x \ne y}} \frac{|D^\alpha \psi(x) - D^\alpha \psi(y)|}{\|x - y\|_2^{d-[d]}} \le c \right\}.$$

When $0 < d \le 1$, we have $[d] = 0$. In this case, the above condition reduces to the usual Hölder continuity condition:

$$\sup_{x,y \in \mathcal{X}, \, x \ne y} \frac{|\psi(x) - \psi(y)|}{\|x - y\|_2^d} \le c.$$

Thus, the notion of $d$-smoothness reduces to Hölder continuity when $0 < d \le 1$.

Since the nonlinear component $f(\cdot)$ belongs to a Hölder class $\Lambda(d, c)$, it admits a linear sieve approximation over the space

$$\mathcal{S}_L = \left\{ x \mapsto \Psi(x)^\top \mathbf{b} : \mathbf{b} \in \mathbb{R}^L \right\},$$

where

$$\Psi(x) = (\psi_1(x), \ldots, \psi_L(x))^\top$$

collects $L$ scalar basis functions on $\mathcal{X}$. In particular, under standard regularity conditions on the sieve basis, there exists a coefficient vector $\mathbf{b} \in \mathbb{R}^L$ such that

$$\inf_{g \in \mathcal{S}_L} \sup_{x \in \mathcal{X}} |f(x) - g(x)| = \sup_{x \in \mathcal{X}} |f(x) - \Psi(x)^\top \mathbf{b}| = O(L^{-d/p}).$$

Moreover, since $f(\cdot)$ is uniformly bounded by $\eta_f^2$ and the sieve basis functions are assumed to be uniformly bounded, then the associated coefficient vector $\mathbf{b}$ can be taken to satisfy a norm bound. Specifically, there exists a constant $\eta_0 > 0$ such that

$$\frac{1}{L} \|\mathbf{b}\|_2^2 = \frac{1}{L} \sum_{l=1}^L b_l^2 \le \eta_0^2, \ \mathbf{b} = (b_1, \cdots, b_L)^\top.$$

Let $\eta = \eta_0 \sqrt{L}$, this bound can be directly incorporated into our derivations, which in turn implies that the misspecification bias is bounded, as discussed after Assumption A.3.

## A.2. Robust Design in Contextual Bandits with Interactions

In this subsection, we move beyond additive treatment effects and consider settings in which treatment effects interact with covariates. In such environments, the impact of an assignment depends on which units are treated, so assignment decisions directly affect covariate-dependent heterogeneity and the information content of the experiment.

We adopt the following working linear model with treatment-covariate interactions, $m(a, X) := \mathbb{E}(Y \mid a, X) \approx a X^\top \xi + X^\top \beta$, and define $\theta = (\xi^\top, \beta^\top)^\top$ as the best linear approximation to the true conditional mean in the least–squares sense,

$$\theta = \arg \min_{\alpha \in \mathbb{R}^{2p}} \sum_{a \in \{-1,1\}} \mathbb{E}\big[m(a, X) - (aX^\top, X^\top)\alpha\big]^2,$$

where the expectation is taken with respect to the distribution of $X$, while the action is averaged uniformly over $\{-1,1\}$. Let $f(a, X) = m(a, X) - (aX^\top, X^\top)\theta$ denote the approximation error. We assume that treatment–covariate interactions are correctly specified in the working model, so that any remaining misspecification depends only on the covariates, i.e., $f(a, X) = f(X)$. The model is

$$Y_i = A_i X_i^\top \xi + X_i^\top \beta + f(X_i) + e_i. \tag{A.8}$$

As in Section 3, this projection definition implies a population-level orthogonality condition, $\mathbb{E}[Xf(X)] = 0$. This orthogonality serves the same purpose as in the additive model.

With treatment-covariate interactions, treatment effects are heterogeneous across covariates. The ATE can be written as a linear functional of $\theta$, ATE $= \mathbf{u}^\top \theta$, $\mathbf{u} = (2\mathbb{E}X^\top, \mathbf{0}^\top)^\top$, where, without loss of generality, covariates are standardized so that $\mathbb{E}X = (1, 0, \ldots, 0)^\top$. We estimate $\theta$ by OLS under the working interaction model and form the plug-in estimator $\widehat{\text{ATE}} = \mathbf{u}^\top \widehat{\theta}$. Conditioning on a fixed assignment sequence $\mathbf{a} \in \{\pm 1\}^N$, the MSE admits a decomposition analogous to that of the robust design without interactions, namely

$$\text{MSE}(\mathbf{u}^\top \widehat{\theta}_\mathbf{a}) = \underbrace{2\sigma^2 \, \mathbb{E}X^\top (G_1^{-1} + G_2^{-1})\mathbb{E}X}_{I_1} + \underbrace{\{\mathbb{E}X^\top [(G_1^{-1} - G_2^{-1})\mathbf{X}^\top + (G_1^{-1} + G_2^{-1})\mathbf{X}^\top D_\mathbf{a}]\boldsymbol{f}\}^2}_{I_2}, \tag{A.9}$$

where

$$G_1 = \sum_{i=1}^N (1+a_i)X_i X_i^\top, \quad G_2 = \sum_{i=1}^N (1-a_i)X_i X_i^\top, \quad D_\mathbf{a} = \text{diag}(a_1, \ldots, a_N), \quad \boldsymbol{f} := f(\mathbf{X}) = (f(X_1), \cdots, f(X_N))^\top.$$

The first term in (A.9) corresponds to the variance arising from noise, while the second term represents the squared bias induced by model misspecification through the nonlinear component $f(\cdot)$. A detailed derivation is provided below.

**Derivation of conditional MSE** (A.9). For a fixed treatment allocation scheme $\mathbf{a} \in \{\pm 1\}^N$, the OLS estimator admits the closed-form expression

$$\widehat{\theta}_\mathbf{a} = (\mathbf{Z}^\top \mathbf{Z})^{-1}\mathbf{Z}^\top \mathbf{Y},$$

where $\mathbf{Z} = [D_\mathbf{a}\mathbf{X}, \mathbf{X}] \in \mathbb{R}^{N \times 2p}$ has $i$th row $(a_i X_i^\top, X_i^\top)$. The conditional MSE takes the same form as in (A.1) with $\mathbf{u} = (2\mathbb{E}X^\top, \mathbf{0}^\top)^\top$. A direct calculation yields

$$(\mathbf{Z}^\top \mathbf{Z})^{-1} = \begin{pmatrix} \mathbf{X}^\top \mathbf{X} & \mathbf{X}^\top D_\mathbf{a} \mathbf{X} \\ \mathbf{X}^\top D_\mathbf{a} \mathbf{X} & \mathbf{X}^\top \mathbf{X} \end{pmatrix}^{-1} = \frac{1}{2} \begin{pmatrix} G_1^{-1} + G_2^{-1} & G_1^{-1} - G_2^{-1} \\ G_1^{-1} - G_2^{-1} & G_1^{-1} + G_2^{-1} \end{pmatrix}.$$

Consequently, the variance component of the MSE satisfies

$$I_1 = \sigma^2 \, \mathbf{u}^\top (\mathbf{Z}^\top \mathbf{Z})^{-1} \mathbf{u} = 2\sigma^2 \, \mathbb{E}X^\top (G_1^{-1} + G_2^{-1})\mathbb{E}X.$$

Similarly, the bias component can be written as

$$I_2 = \mathbf{u}^\top (\mathbf{Z}^\top \mathbf{Z})^{-1}\mathbf{Z}^\top (\boldsymbol{f}\boldsymbol{f}^\top)\mathbf{Z}(\mathbf{Z}^\top \mathbf{Z})^{-1}\mathbf{u} = \left\{ \mathbb{E}X^\top [(G_1^{-1} + G_2^{-1})\mathbf{X}^\top D_\mathbf{a} + (G_1^{-1} - G_2^{-1})\mathbf{X}^\top]\boldsymbol{f} \right\}^2.$$

Combining the expressions for $I_1$ and $I_2$ yields the conditional MSE representation in (A.9).

**Derivation of an upper bound for the MSE.** Similar to the derivation of (9), we first establish an upper bound in Proposition A.2 and then construct a surrogate version of this upper bound by replacing certain quantities with their population counterparts.

**Proposition A.2.** *Consider the model* (A.8)*, and suppose that the Assumptions A.1–A.3 hold. For any fixed assignment sequence* $\mathbf{a} \in \{\pm 1\}^N$*, the conditional MSE of the ATE estimator admits the robust upper bound, for some constant* $\eta > 0$*:*

$$
\begin{aligned}
\mathrm{MSE}(\widehat{\mathrm{ATE}}) \leq & \frac{2\sigma^2}{N} \mathbb{E}X^\top \Big[ (\Sigma_N + \frac{\Omega_{1,N}}{N})^{-1} + (\Sigma_N - \frac{\Omega_{1,N}}{N})^{-1} \Big] \mathbb{E}X \\
& + \eta^2 \Big\| \mathbf{U}^\top \Big[ (\Sigma_N + \frac{\Omega_{1,N}}{N})^{-1} (\Xi_N^\top + \frac{\Omega_{2,N}}{N}) - (\Sigma_N - \frac{\Omega_{1,N}}{N})^{-1} (\Xi_N^\top - \frac{\Omega_{2,N}}{N}) \Big]^\top \mathbb{E}X \Big\|_2^2 ,
\end{aligned}
\tag{A.10}
$$

*where the matrices* $\Sigma_N, \Xi_N, \mathbf{U}$ *are defined in Proposition A.1, and the second-order imbalance statistics are denoted as*

$$
\Omega_{1,N} = \sum_{i=1}^N a_i X_i X_i^\top, \quad \Omega_{2,N} = \sum_{i=1}^N a_i X_i \Psi^\top(X_i).
$$

*Proof.* We first consider the variance component in (A.9). A direct algebraic manipulation yields

$$
I_1 = 2\sigma^2 \, \mathbb{E}X^\top (G_1^{-1} + G_2^{-1}) \mathbb{E}X = \frac{2\sigma^2}{N} \mathbb{E}X^\top (H_1 + H_2) \mathbb{E}X,
$$

where

$$
H_1 = (\Sigma_N + \Omega_{1,N}/N)^{-1}, \qquad H_2 = (\Sigma_N - \Omega_{1,N}/N)^{-1}.
$$

Next, we turn to the bias component in (A.9).

$$
I_2 = \Big\{ \mathbb{E}X^\top \big[ (G_1^{-1} + G_2^{-1}) \mathbf{X}^\top \mathbf{D_a} + (G_1^{-1} - G_2^{-1}) \mathbf{X}^\top \big] \boldsymbol{f} \Big\}^2 .
$$

By the same sieve-based argument used in the proof of Proposition A.1, the nonlinear component admits the representation

$$
f(\mathbf{X}) = \eta \, \boldsymbol{\Psi}(\mathbf{X}) \mathbf{U} \boldsymbol{\kappa}, \qquad \|\boldsymbol{\kappa}\|_2 \leq 1,
$$

up to a remainder that is absorbed into the worst-case bound. Substituting this expression into $I_2$ yields

$$
I_2 = \eta^2 \Big\{ \boldsymbol{\kappa}^\top \mathbf{U}^\top \big[ H_1 H_3 - H_2 H_4 \big]^\top \mathbb{E}X \Big\}^2 \leq \eta^2 \big\| \mathbf{U}^\top (H_1 H_3 - H_2 H_4)^\top \mathbb{E}X \big\|_2^2 ,
$$

where

$$
H_3 = \Xi_N^\top + \Omega_{2,N}/N, \qquad H_4 = \Xi_N^\top - \Omega_{2,N}/N.
$$

The inequality follows from taking the supremum over all $\boldsymbol{\kappa}$ satisfying $\|\boldsymbol{\kappa}\|_2 \leq 1$. Combining the bounds for $I_1$ and $I_2$ can yield the following conditional MSE upper bound:

$$
\begin{aligned}
\mathrm{MSE}(\widehat{\mathrm{ATE}}) \leq & \frac{2\sigma^2}{N} \mathbb{E}X^\top \Big[ (\Sigma_N + \frac{\Omega_{1,N}}{N})^{-1} + (\Sigma_N - \frac{\Omega_{1,N}}{N})^{-1} \Big] \mathbb{E}X \\
& + \eta^2 \Big\| \mathbf{U}^\top \Big[ (\Sigma_N + \frac{\Omega_{1,N}}{N})^{-1} (\Xi_N^\top + \frac{\Omega_{2,N}}{N}) - (\Sigma_N - \frac{\Omega_{1,N}}{N})^{-1} (\Xi_N^\top - \frac{\Omega_{2,N}}{N}) \Big]^\top \mathbb{E}X \Big\|_2^2 ,
\end{aligned}
$$

The proof is hence completed. $\square$

**Surrogate Objective.** We also invoke a law-of-large-numbers approximation and replace these sample moments with their population counterparts, namely $\Sigma_N \approx \Sigma$ and $\Xi_N \approx \Xi$. Under this approximation, we can obtain a robust approximate upper bound (A.10) by:

$$
\frac{2\sigma^2}{N} \mathbb{E}X^\top (H_1' + H_2') \mathbb{E}X + \eta^2 \big\| \mathbf{U}^\top (H_1' H_3' - H_2' H_4')^\top \mathbb{E}X \big\|_2^2 ,,
\tag{A.11}
$$

where $H_1' = (\Sigma + \Omega_{1,N}/N)^{-1}$, $H_2' = (\Sigma - \Omega_{1,N}/N)^{-1}$, $H_3' = \Xi^\top + \Omega_{2,N}/N$, $H_4' = \Xi^\top - \Omega_{2,N}/N$. This surrogate objective eliminates the dependence on full-sample moments and yields a Markovian state representation that is amenable to dynamic optimization. The explicit Bellman recursion is established in Theorem A.1.

**Theorem A.1.** *Suppose that Assumptions A.1–A.3 hold. Let $\mathcal{U}_N(\Omega_{1,N}, \Omega_{2,N})$ denote the MSE upper bound in (A.11), which serves as our ultimate objective function. Then the optimal sequential design problem can be formulated as a dynamic program with Markov state $(\Omega_{1,i}, \Omega_{2,i})$ and value functions $\{\mathcal{U}_i\}_{i=1}^{N-1}$ defined through the following Bellman equation. At Stage $i$, the value function $\mathcal{U}_i(\Omega_{1,i}, \Omega_{2,i})$ is defined as*

$$\mathcal{U}_i(\Omega_{1,i}, \Omega_{2,i}) = \mathbb{E}\left[ \min_{a \in \{-1,1\}} \mathcal{U}_{i+1}\big(\Omega_{1,i} + aX_{i+1}X_{i+1}^\top, \Omega_{2,i} + aX_{i+1}\Psi^\top(X_{i+1})\big) \right].$$

*Then an optimal design is obtained by choosing, at each stage $i$,*

$$a_i^* \in \arg\min_{a \in \{-1,1\}} \mathcal{U}_i\big(\Omega_{1,i-1} + aX_iX_i^\top, \Omega_{2,i-1} + aX_i\Psi^\top(X_i)\big).$$

*Proof.* We first formulate the sequential design problem as

$$(\text{P2}) := \min_{\mathbf{a}} \quad \mathbb{E}\left[ \frac{2\sigma^2}{N} \mathbb{E}X^\top(H_1' + H_2')\mathbb{E}X + \eta^2 \big\| \mathbf{U}^\top(H_1'H_3' - H_2'H_4')^\top \mathbb{E}X \big\|_2^2 \right]$$

$$\text{s.t.} \quad a_i \in \{\pm 1\}, \quad a_i \text{ is } \mathcal{F}_i\text{-measurable}, \quad i = 1, \ldots, N.$$

Then we formulate problem (P2) as a finite-horizon MDP, following the same construction as in the proof of Theorem 1.

At decision stage $i$, the state is defined as

$$S_i = (\Omega_{1,i-1}, \Omega_{2,i-1}, X_i),$$

with terminal state $S_{N+1} = (\Omega_{1,N}, \Omega_{2,N})$. The action space at each stage is $\{-1, 1\}$. If action $a_i$ is chosen at state $S_i$, the state transitions according to

$$(\Omega_{1,i-1}, \Omega_{2,i-1}, X_i) \longrightarrow (\Omega_{1,i-1} + a_iX_iX_i^\top, \Omega_{2,i-1} + a_iX_i\Psi^\top(X_i), X_{i+1}).$$

There is no per-stage loss, and the terminal objective is given by

$$(\Omega_{1,N}, \Omega_{2,N}) \longmapsto \mathcal{U}_N(\Omega_{1,N}, \Omega_{2,N}).$$

Since the horizon is finite and the action space is finite, the problem is a terminal-loss MDP. By Proposition 4.2.1 in Bertsekas (2022), there exists an optimal policy and a sequence of value functions $\{J_i^*\}_{i=1}^{N+1}$ satisfying

$$J_{N+1}^*(\Omega_{1,N}, \Omega_{2,N}) = \mathcal{U}_N(\Omega_{1,N}, \Omega_{2,N}),$$

and the Bellman recursion

$$J_i^*(\Omega_{1,i-1}, \Omega_{2,i-1}, X_i) = \min_{a \in \{-1,1\}} \mathbb{E}\big[ J_{i+1}^*\big(\Omega_{1,i-1} + aX_iX_i^\top, \Omega_{2,i-1} + aX_i\Psi^\top(X_i), X_{i+1}\big) \big]. \tag{A.12}$$

Define the reduced value functions

$$\mathcal{U}_i(\Omega_{1,i}, \Omega_{2,i}) = \mathbb{E}\big[ J_{i+1}^*(\Omega_{1,i}, \Omega_{2,i}, X_{i+1}) \big].$$

Then (A.12) implies

$$\mathcal{U}_i(\Omega_{1,i}, \Omega_{2,i}) = \mathbb{E}\left[ \min_{a \in \{-1,1\}} \mathcal{U}_{i+1}\big(\Omega_{1,i} + aX_{i+1}X_{i+1}^\top, \Omega_{2,i} + aX_{i+1}\Psi^\top(X_{i+1})\big) \right],$$

and the optimal policy satisfies

$$a_i^* \in \arg\min_{a \in \{-1,1\}} \mathcal{U}_i\big(\Omega_{1,i-1} + aX_iX_i^\top, \Omega_{2,i-1} + aX_i\Psi^\top(X_i)\big).$$

This establishes the Bellman recursion and completes the proof. $\square$

## A.3. Robust Sequential Design in Dynamic Settings

In this subsection, we consider the dynamic setting and begin by describing the setup. Unlike contextual bandits, the covariate process $X_{it}$ is endogenous in dynamic settings. Through the state dynamics, the distribution of covariates at each time step depends on the experimental policy $\pi$ itself. As a result, the best linear approximation to the conditional mean at each time step is inherently policy-dependent.

Formally, for a given experimental policy $\pi$, define the time-$t$ projection parameter

$$\theta_t^\pi = \arg \min_{\alpha \in \mathbb{R}^{2p}} \mathbb{E}_{A_t, X_t} \left[ \mathbb{E}(Y_t \mid A_t, X_t) - (A_t X_t^\top, X_t^\top)\alpha \right]^2,$$

where the expectation is taken under the joint distribution of $(A_t, X_t)$ induced by the policy $\pi$. Let $f_t^\pi(A, X) = \mathbb{E}(Y_t \mid A_t, X_t) - (A_t X_t^\top, X_t^\top)\theta_t^\pi$ denote the corresponding approximation error. As in Section 3.2, we assume that $f_t^\pi(A, X) = f_t^\pi(X)$, which implies the orthogonality condition

$$\mathbb{E}\big[X_t f_t^\pi(X_t)\big] = 0.$$

This policy dependence is intrinsic to dynamic experiments: the statistical target $\theta_t^\pi$ itself varies with the policy that generates the data. When the analyst estimates the ATE via OLS using data generated under an allocation policy $\pi$, the resulting estimand is $\theta_t^\pi$, when the true outcome model is nonlinear. From this perspective, we define

$$\text{ATE}(\pi) = \frac{1}{T} \sum_{t=1}^{T} \mathbf{u}_t^\top \theta_t^\pi$$

as the appropriate target when data are collected under policy $\pi$. Accordingly, we adopt a plug-in estimator $\widehat{\text{ATE}}(\pi)$, where $\mathbf{u}_t$ is estimated from the dynamic covariate system and $\theta_t^\pi$ is estimated via OLS.

If the true causal estimand ATE defined in Section 4 deviates substantially from the working estimand $\text{ATE}(\pi)$, consistent estimation would generally require assigning all time points in a day persistently to a single treatment arm and estimating treatment effects separately across groups. In this paper, we instead focus on a regime in which the discrepancy between $\text{ATE}(\pi)$ and ATE is dominated by the statistical estimation error of $\widehat{\text{ATE}}(\pi)$. This regime is particularly relevant when treatment effects are relatively small, and the primary source of uncertainty arises from finite-sample estimation rather than policy-induced target drift. Under this perspective, the role of experimental design is to select a policy $\pi$ that induces favorable covariate trajectories and treatment allocations, thereby controlling a robust upper bound on $\text{MSE}(\widehat{\text{ATE}}(\pi))$.

In what follows, we first state the required assumptions and derive an upper bound on the MSE. We then prove the across-day and within-day Bellman recursions in Section 4.

**Assumption A.4.** (Noise independence) The noise variables $\{e_{i,t}\}$ are independent across units $i$ and independent over time. In particular, for any $i$ and $t_1, t_2 \in \{1, \ldots, T\}$, $\text{Cov}(e_{i,t_1}, e_{i,t_2}) = \sigma_{t_1}^2 \mathbb{I}(t_1 = t_2)$.

**Assumption A.5.** (Non-singular covariance matrix) For the policy $\pi$ of interest, the covariance matrix $\mathbb{E}^\pi[\mathbf{X}_t^\top \mathbf{X}_t]$ is positive definite for all $t = 1, \ldots, T$.

**Assumption A.6.** (Hölder smoothness & bounded sieves) For any policy $\pi$ and each $t = 1, \ldots, T$, the nonlinear component $f_t^\pi(\cdot)$ belongs to a Hölder class $\Lambda(d, c)$ defined in Appendix A.1. Moreover, the associated sieve basis functions are uniformly bounded, and $f_t^\pi(\cdot)$ is uniformly bounded in the sense that $\sup_x |f_t^\pi(x)| \leq \eta_{f_t}^2$ for some constant $\eta_{f_t} > 0$.

**Assumption A.7.** (Weak signal) For the policy $\pi$ of interest, there exist some small constants $\delta_1 > 0$ and $\delta_2 > 0$ such that $\max_{1 \leq t \leq T} \|\beta_t^\pi\|_2 \leq \delta_1$ and $\max_{1 \leq t \leq T} \|\xi_t^\pi\|_2 \leq \delta_2$.

Assumptions A.4–A.6 are the dynamic versions of Assumptions A.1–A.3. As discussed in the introduction, the weak-signal condition is common in ride-sharing platform settings (Tang et al., 2019; Sun et al., 2024).

**Derivation of an upper bound for the MSE.** To illustrate the main procedure, we proceed in two steps. We first derive the conditional MSE of the ATE under a correctly specified linear model and then extend the analysis to the misspecified setting.

**Step 1: Derivation of conditional MSE under the linear model**. If the linear model is correctly specified, the parameter $\theta_t$ does not depend on the experimental policy. Fix a treatment sequence $\{a_{it}\}$ from the policy $\pi$. For each time $t$, the OLS

estimator admits the closed-form expression

$$\widehat{\theta}_t^\pi = \left( \frac{1}{N} \sum_{i=1}^N Z_{it} Z_{it}^\top \right)^{-1} \frac{1}{N} \sum_{i=1}^N Z_{it} Y_{it},$$

where $Z_{it} = (a_{it} X_{it}^\top, X_{it}^\top)^\top \in \mathbb{R}^{2p}$. The plug-in ATE estimator is $\widehat{\text{ATE}} = T^{-1} \sum_{t=1}^T \widehat{\mathbf{u}}_t^\top \widehat{\theta}_t^\pi$. Then we have the following decomposition:

$$\widehat{\text{ATE}} - \text{ATE} = \frac{1}{T} \sum_{t=1}^T \left[ \mathbf{u}_t^\top (\widehat{\theta}_t^\pi - \theta_t) + \theta_t^\top (\widehat{\mathbf{u}}_t - \mathbf{u}_t) + (\widehat{\mathbf{u}}_t - \mathbf{u}_t)^\top (\widehat{\theta}_t^\pi - \theta_t) \right].$$

Under the common weak-signal condition (Sun et al., 2024; Farias et al., 2022; Tang et al., 2019)-formalized in Assumption A.7-the second-order remainder is of higher order and can be neglected. Consequently, under the linear model with the OLS estimator, the MSE of $\widehat{\text{ATE}}$ reduces to its variance term

$$\text{MSE} = \text{Var}(\widehat{\text{ATE}}) = \text{Var}\left( \frac{1}{T} \sum_{t=1}^T \mathbf{u}_t^\top \widehat{\theta}_t^\pi \right),$$

where $\mathbf{u}_t = (\mathbb{E}^{(1)} X_t^\top + \mathbb{E}^{(-1)} X_t^\top, \mathbb{E}^{(1)} X_t^\top - \mathbb{E}^{(-1)} X_t^\top)^\top$. A direct calculation yields the decomposition

$$\text{Var}(\widehat{\text{ATE}}) = \frac{1}{T^2} \sum_{t_1=1}^T \sum_{t_2=1}^T \mathbf{u}_{t_1}^\top \left[ \frac{1}{N} \sum_{i=1}^N Z_{i,t_1} Z_{i,t_1}^\top \right]^{-1} \left( \frac{1}{N^2} \sum_{i=1}^N Z_{i,t_1} Z_{i,t_2}^\top \sigma_e^2(t_1, t_2) \right) \left[ \frac{1}{N} \sum_{i=1}^N Z_{i,t_2} Z_{i,t_2}^\top \right]^{-1} \mathbf{u}_{t_2}.$$

Under Assumption A.4, the cross-time covariance vanishes, and the variance term simplifies to

$$\frac{1}{NT^2} \sum_{t=1}^T \mathbf{u}_t^\top G_t^{-1} \mathbf{u}_t \, \sigma_t^2,$$

where $G_t = \frac{1}{N} \sum_{i=1}^N Z_{it} Z_{it}^\top$ and $\sigma_t^2 = \text{Var}(e_t)$. Consequently, we can obtain the exact conditional MSE under the linear model.

**Step 2: Derivation of an upper bound** (13) **for the MSE under the model misspecification**. Note that the plug-in estimator is given by $\widehat{\text{ATE}}(\pi) = T^{-1} \sum_{t=1}^T \widehat{\mathbf{u}}_t \widehat{\theta}_t^\pi$. Under the Assumptions A.4 and A.7, the MSE of $\widehat{\text{ATE}}(\pi)$ is given by

$$\text{MSE}(\widehat{\text{ATE}}(\pi)) = \frac{1}{NT^2} \sum_{t=1}^T \mathbf{u}_t^\top G_t^{-1} \mathbf{u}_t \, \sigma_t^2 + \left( \frac{1}{T} \sum_{t=1}^T \mathbf{u}_t^\top G_t^{-1} \frac{1}{N} \sum_{i=1}^N Z_{it} f_t^\pi(X_{it}) \right)^2. \tag{A.13}$$

Next, we bound the bias term on the right-hand side of (A.13). By Assumption A.6, together with the Cauchy–Schwarz inequality and the linear sieve approximation $f_t^\pi(x) \approx (\mathbf{b}_t^\pi)^\top \Psi(x)$, we obtain

The second term on the right-hand side of (A.13)

$$\leq \frac{1}{T} \sum_{t=1}^T \left[ \frac{1}{N} \sum_{i=1}^N Z_{it}^\top f_t^\pi(X_{it}) \right] G_t^{-1} \mathbf{u}_t \mathbf{u}_t^\top G_t^{-1} \left[ \frac{1}{N} \sum_{i=1}^N Z_{it} f_t^\pi(X_{it}) \right]$$

$$\approx \frac{1}{T} \sum_{t=1}^T (\mathbf{b}_t^\pi)^\top \underbrace{\left[ \frac{1}{N} \sum_{i=1}^N \Psi(X_{it}) Z_{it}^\top \right]}_{H_t} G_t^{-1} \mathbf{u}_t \mathbf{u}_t^\top G_t^{-1} \underbrace{\left[ \frac{1}{N} \sum_{i=1}^N Z_{it} \Psi^\top(X_{it}) \right]}_{H_t^\top} \mathbf{b}_t^\pi,$$

where $H_t := N^{-1} \sum_{i=1}^N \Psi(X_{it}) Z_{it}^\top$. Moreover, by the orthogonality condition $\mathbb{E}_{X_t}[X_t f_t^\pi(X_t)] = 0$, we have

$$\mathbb{E}_{X_t} \left[ \frac{1}{N} \sum_{i=1}^N X_{it} f_t^\pi(X_{it}) \right] = 0.$$

Define

$$\mathbf{\Psi}_t = \begin{pmatrix} \Psi(X_{1t})^\top \\ \vdots \\ \Psi(X_{Nt})^\top \end{pmatrix} \in \mathbb{R}^{N \times L}, \qquad \mathbf{X}_t = \begin{pmatrix} X_{1t}^\top \\ \vdots \\ X_{Nt}^\top \end{pmatrix} \in \mathbb{R}^{N \times p},$$

and then

$$\mathbf{C}_t^\pi := \mathbb{E}_{X_t}\left[ \frac{1}{N} \mathbf{\Psi}_t^\top \mathbf{X}_t \right] \in \mathbb{R}^{L \times p}.$$

Following the orthogonality analysis in Section 3.2, and under Assumption A.6, the sieve coefficient vector admits the representation $\mathbf{b}_t^\pi \approx \eta_t \mathbf{U}_t^\pi \boldsymbol{\kappa}_t^\pi$, with $\|\boldsymbol{\kappa}_t^\pi\|_2 \leq 1$. For simplicity, we suppress the superscript $\pi$ in what follows. Consequently,

$$\text{The second term on the right-hand side of (A.13)} \leq \frac{1}{T} \sum_{t=1}^T \eta_t^2 \big\| \mathbf{U}_t^\top H_t G_t^{-1} \mathbf{u}_t \big\|_2^2.$$

Substituting the upper bound into (A.13) yields (13).

**Proof of Theorem 2**. To establish the across-day Bellman recursion, we first formulate the robust design problem as a day-level, time-dependent MDP with a finite horizon. We then invoke the Bellman optimality equations for time-dependent MDPs to complete the proof.

*Proof.* We view the across-day design problem as a finite-horizon, time-dependent MDP indexed by days $i = 0, 1, \ldots, N$.

**State.** The day-level state at the end of day $i$ is $\mathcal{S}_i = \{(B_{it}, C_{it}, D_{it}, F_{it})\}_{t=1}^T$, which aggregates all cross-day sufficient statistics required by the terminal objective $\mathcal{Q}_N(\mathcal{S}_N)$, where $(B_{it}, C_{it}, D_{it}, F_{it})$ is defined in Section 4 and $\mathcal{Q}_N(\mathcal{S}_N)$ is the MSE upper bound in (13).

**Action.** At the beginning of day $i+1$, the decision maker chooses a within-day policy $\pi \in \Pi$, where $\pi = \{\pi_t\}_{t=1}^T$ and each $\pi_t$ maps $(\mathcal{S}_i, \mathcal{H}_{i+1,t})$ to $\{\pm 1\}$, where $\mathcal{H}_{i,t} = \{X_{i,1:t}, A_{i,1:(t-1)}\}$ denotes the information available at time $t$ within day $i$. Conditional on $(\mathcal{S}_i, \pi)$ and $\mathcal{H}_{i+1,t}$, the within-day actions $A_{i+1,t}$ are generated sequentially by $\pi$.

**Transition.** Given a realized covariate trajectory $X_{i+1,1:T}$ and a policy $\pi \in \Pi$, define the one-day increment $\Phi(X_{i+1,1:T}, \pi)$:

$$\{(X_{i+1,t}X_{i+1,t}^\top, A_{i+1,t}X_{i+1,t}X_{i+1,t}^\top, \Psi(X_{i+1,t})X_{i+1,t}^\top, A_{i+1,t}\Psi(X_{i+1,t})X_{i+1,t}^\top)\}_{t=1}^T,$$

and $A_{i+1,t}$ is generated sequentially by $\pi$ based on $(\mathcal{S}_i, \mathcal{H}_{i+1,t})$. The state evolves as $\mathcal{S}_{i+1} = \mathcal{S}_i \oplus \Phi(X_{i+1,1:T}, \pi)$, where $\oplus$ denotes component-wise addition. Under the assumed across-day sampling scheme for $X_{i+1,1:T}$, the induced transition kernel of $\mathcal{S}_{i+1}$ depends only on $(\mathcal{S}_i, \pi)$, hence $\{\mathcal{S}_i\}$ is Markov at the day level.

**Terminal reward.** There is no per step reward. The terminal reward is $\mathcal{Q}_N(\mathcal{S}_N)$.

Given a realized covariate trajectory $X_{i,1:T}$ and a policy $\pi \in \Pi$, define the one-day increment $\Phi(X_{i,1:T}, \pi)$:

$$\{(X_{it}X_{it}^\top, A_{it}X_{it}X_{it}^\top, \Psi(X_{it})X_{it}^\top, A_{it}\Psi(X_{it})X_{it}^\top)\}_{t=1}^T,$$

and $A_{it}$ is generated sequentially by $\pi$ based on $(\mathcal{S}_{i-1,t}, \mathcal{H}_{i,t})$. Therefore, by the Bellman optimality principle for finite-horizon (time-dependent) MDPs, see the Theorem 1.9 in Agarwal et al. (2019), the value functions $\{\mathcal{Q}_i\}_{i=0}^{N-1}$ satisfy

$$\mathcal{Q}_i(\mathcal{S}_i) = \mathbb{E}\Big[ \inf_{\pi \in \Pi} \mathcal{Q}_{i+1}\big(\mathcal{S}_i \oplus \Phi(X_{i+1,1:T}, \pi)\big) \Big], \quad i = N-1, \ldots, 0,$$

and there exists an optimal policy sequence $\{\pi_i^\star\}_{i=1}^N$ attaining the infimum at each state. This establishes the Bellman recursion and completes the proof. $\qquad \square$

**Proof of Theorem 3**. Based on the formulation of **Within-day RL** in Section 4, we can similarly define a finite-horizon MDP for within-day data generation. The result then follows by an argument analogous to the proof of Theorem 2, and we omit the proof for brevity.

# B. Implementation Details.

In this section, we provide the implementation details for the main text, including all algorithms and their corresponding explanations.

## B.1. Details for Algorithms 1 and 2

To summarize our carefully designed procedure for contextual bandits without interactions, we make the following remarks on Algorithms 1 and 2.

- **The behavior policy** $\pi_b$**.** In the 5th Line of Algorithm1, our goal is to ensure sufficient coverage of the state space of the imbalance statistics $(\Delta_n, \Gamma_n)$, so that the Q-function can be accurately approximated over the relevant region. In principle, for each synthetic sample $b$, one may consider all $n + 1$ action sequences of length $n$ that contain exactly $k \in \{0, \ldots, n\}$ entries equal to $+1$ and $n - k$ entries equal to $-1$. Under i.i.d. covariates and exchangeability of action sequences, these configurations fully characterize the range of the action-imbalance statistic

$$\sum_{i=1}^{n} a_i = 2k - n.$$

  Thus, the total number of distinct realizations in this step is of order $\mathcal{O}(nB)$. For simplicity of presentation, Algorithm 1 constructs only $B$ representative samples. For each such sample, the associated imbalance statistics $\Delta_n^{(b)}$ and $\Gamma_n^{(b)}$ are computed based on the corresponding action sequence. In fact, for each synthetic sample $b$, we implicitly consider all action assignments with different imbalance levels indexed by $k = 0, \ldots, n$, in order to enrich the variability of the training samples. Overall, to ensure sufficient exploration of the representation space, we use two strategies. First, the behavior policy $\pi_b$ is designed to span a wide range of imbalance levels. Second, given $\pi_b$, we generate $B$ independent synthetic rollouts. As $B$ increases, the resulting dataset covers a richer set of $(\Delta_n, \Gamma_n)$ values. In our experiments, we set $B$ on the order of thousands to ensure adequate exploration.

- **Monte Carlo average.** To approximate the expectation in (A.7), during the training of the $n$-th Q-value network $Q_n$ for $1 \leq n < N$, we draw a shared set of samples $\{X_{n+1}^{(m)}\}_{m=1}^{M}$ from $\mathcal{F}_X$.

- **The calculation of the terminal value function.** These priors $(\Sigma, \Xi, \mathbf{U}, \nu^2 = \eta^2/\sigma^2)$ are necessary when $n = N - 1$, because the objective function can then be rewritten explicitly as

$$Q_N(\Delta_N, \Gamma_N) \;\propto\; \frac{1}{N - \frac{1}{N}\Delta_N^\top \Sigma^{-1} \Delta_N} + \frac{\nu^2 \left\| \mathbf{U}^\top (\Gamma_N - \Xi \Sigma^{-1} \Delta_N) \right\|_2^2}{\left( N - \frac{1}{N}\Delta_N^\top \Sigma^{-1} \Delta_N \right)^2}.$$

  A larger $\nu^2$ assigns more weight to the impact of model misspecification, and vice versa. Equivalently, $\nu^2$ can be interpreted as a robustness (penalty) parameter. The matrices $\Sigma$, $\mathbf{U}$, and $\Xi$ can be pre-estimated from a large collection of empirical covariate observations.

- **Recursive training procedure.** We train the Q-networks backward in $n$: training $Q_{n-1}$ relies on the already trained $Q_n$. During training, extreme action allocations can lead to fitted Q-values with substantial outliers and unusually large magnitudes. To stabilize learning, we exploit the monotonicity of the logarithm and its tendency to produce more Gaussian-like targets. Starting from stage $N - 1$ and proceeding recursively, we apply a log transformation to the original Q-function targets and train the Q-networks on the transformed values, which improves the quality and stability of the fit.

- **Solving for Optimal Actions**. After collecting the fitted Q value-networks $\{Q_n\}_{n=1}^{N-1} \cup Q_N$ by the Algorithm 1, it can be used to dynamically make decisions for each day via Algorithm 2. For $n = 1, \cdots, N - 1$, we have

$$a_n^* \in \underset{a \in \{-1,1\}}{\arg\min} \; Q_n(\Delta_n(\mathbf{a}_{1:(n-1)}^*, a), \Gamma_n(\mathbf{a}_{1:(n-1)}^*, a)),$$

  where $\Delta_n(\mathbf{a}_{1:(n-1)}^*, a) = \sum_{k=1}^{n-1} a_k X_k + a X_n$, $\Gamma_n(\mathbf{a}_{1:(n-1)}^*, a) = \sum_{k=1}^{n-1} a_k^* \Psi(X_k) + a \Psi(X_n)$, if $n = 1$, then $\mathbf{a}_{1:0}^* = 0$. And

$$a_N^* \in \underset{a \in \{-1,1\}}{\arg\min} \; Q_N(\Delta_N(\mathbf{a}_{1:(N-1)}^*, a), \Gamma_N(\mathbf{a}_{1:(N-1)}^*, a)).$$

  Therefore, we can collect a sequence of data under the proposed optimal decision mechanism, $\{X_n, a_n^*, Y_n\}_{n=1}^{N}$, which then can be used to estimate ATE via OLS estimate.

## B.2. Algorithm for Contextual Bandits with Interactions

We summarize the proposed procedure for handling treatment–covariate interactions in Algorithm 3. We begin by defining the key imbalance statistics used in the model:

$$\Omega_{1,n} = \sum_{i=1}^{n} a_i X_i X_i^{\top}, \ \Omega_{2,n} = \sum_{i=1}^{n} a_i X_i \Psi^{\top}(X_i). \tag{A.14}$$

---

**Algorithm 3** Training $\mathcal{U}_n$ via Synthetic Rollouts

---

1: **Input:** A covariate distribution $\mathcal{F}_X$ (or its empirical version using historical data), a behavior policy $\pi_b$, number of rollouts $B$, value function $\mathcal{U}_{n+1}$.
2: Sample covariates $\{X_{n+1}^{(m)}\}_{m=1}^{M}$ from $\mathcal{F}_X$.
3: **for** $b = 1, \ldots, B$ **do**
4:      Sample $\{X_i^{(b)}\}_{i=1}^{n}$ from $\mathcal{F}_X$.
5:      Sample $\{a_i^{(b)}\}_{i=1}^{n} \in \{-1, 1\}^n$ from $\pi_b$.
6:      Calculate $\Omega_{1,n}^{(b)}$, $\Omega_{2,n}^{(b)}$ by (A.14).
7:      Calculate the target by Monte Carlo averaging:

$$\mathcal{J}_n^{(b)} = \frac{1}{M} \sum_{m=1}^{M} \min_{a \in \{-1,1\}} \mathcal{U}_{n+1}\Big( \Omega_{1,n}^{(b)} + a X_{n+1}^{(m)}(X_{n+1}^{(m)})^{\top}, \ \Omega_{2,n}^{(b)} + a X_{n+1}^{(m)} \Psi^{\top}(X_{n+1}^{(m)}) \Big).$$

8: **end for**
9: Fit a DNN regressor on $\{(\Omega_{1,n}^{(b)}, \Omega_{2,n}^{(b)}, \mathcal{J}_n^{(b)})\}_{b=1}^{B}$ to obtain $\mathcal{U}_n$.
10: **Output:** $\mathcal{U}_n$.

---

The main procedure of Algorithm 3 is very similar to that of Algorithm 1. We emphasize the following differences:

- **Objective.** In Algorithm 3, the $N$-step objective function $\mathcal{U}_N$ is given exactly by

$$\mathcal{U}_N\big(\Omega_{1,N}, \Omega_{2,N}\big) \propto \frac{2}{N} (\mathbb{E}X)^{\top} \big(H_1' + H_2'\big)(\mathbb{E}X) \ + \ \nu^2 \Big\| \mathbf{U}^{\top}\big(H_1' H_3' - H_2' H_4'\big)^{\top} \mathbb{E}X \Big\|_2^2,$$

where

$$H_1' = \Big(\Sigma + \frac{\Omega_{1,N}}{N}\Big)^{-1}, \quad H_2' = \Big(\Sigma - \frac{\Omega_{1,N}}{N}\Big)^{-1}, \quad H_3' = \Xi^{\top} + \frac{\Omega_{2,N}}{N}, \quad H_4' = \Xi^{\top} - \frac{\Omega_{2,N}}{N}.$$

  As noted earlier, this objective involves the second-order imbalance statistics rather than the first-order imbalance statistics.

- **The dimensions of inputs**. Because the inputs $(\Omega_{1,n}, \Omega_{2,n})$ are $(p \times p)$, $(p \times L)$ matrices, respectively. Although there exist neural networks designed for matrix-valued inputs that can improve data efficiency, for simplicity we vectorize these matrices and use each entry as an input feature. Consequently, the input dimension is $p^2 + pL$.

## B.3. Algorithm for Dynamic Settings

**Implementation Details of the Hierarchical Design Framework.** To clarify the proposed **RSD** framework, which integrates day-level dynamic programming with within-day reinforcement learning, we present the corresponding pseudocode in Algorithms 4 and 5.

For clarity, we first present the policy learning algorithm for the $N$-th day, which isolates and illustrates the within-day reinforcement learning procedure. We then describe the policy learning algorithm for an arbitrary $n$th day $(n < N)$, highlighting how dynamic programming is performed across days.

**Algorithm for the last day.** Consider the decision-making problem at the first time point of the $N$-th day. By this time, the entire history from the previous $N - 1$ days has been observed. Importantly, however, the resulting MSE depends on the

past only through a collection of summary statistics, rather than the full trajectory history. Specifically, for each time period $1 \leq t \leq T$, define

$$\mathcal{S}_{N-1,t} := \{\Sigma_{N-1}^{(t)}, \Omega_{1,N-1}^{(t)}, \Xi_{N-1}^{(t)}, \Omega_{2,N-1}^{(t)}\}.$$

These quantities are given by

$$\Sigma_{N-1}^{(t)} = \frac{1}{N}\sum_{i=1}^{N-1} X_{it}X_{it}^{\top}, \qquad \Omega_{1,N-1}^{(t)} = \frac{1}{N}\sum_{i=1}^{N-1} A_{it}X_{it}X_{it}^{\top},$$

$$\Xi_{N-1}^{(t)} = \frac{1}{N}\sum_{i=1}^{N-1} \Psi(X_{it})X_{it}^{\top}, \qquad \Omega_{2,N-1}^{(t)} = \frac{1}{N}\sum_{i=1}^{N-1} A_{it}\Psi(X_{it})X_{it}^{\top}.$$

Given these summary statistics, once the data $(X_{Nt}, A_{Nt})$ at time $t$ on the $N$-th day are observed, we can evaluate the contribution of time period $t$ to the overall objective across all days. We define this contribution as the *immediate reward*

$$R_{N,t} = -\frac{1}{T}\left\{\frac{\sigma_t^2}{N}\mathbf{u}_t^{\top}G_t^{-1}\mathbf{u}_t + \eta_t^2\|\mathbf{U}_t^{\top}H_tG_t^{-1}\mathbf{u}_t\|_2^2\right\}. \tag{A.15}$$

Here,

$$G_t = \frac{1}{N}\sum_{i=1}^{N} Z_{it}Z_{it}^{\top} = \begin{pmatrix} \Sigma_{N-1}^{(t)} + \frac{1}{N}X_{Nt}X_{Nt}^{\top} & \Omega_{1,N-1}^{(t)} + \frac{1}{N}A_{Nt}X_{Nt}X_{Nt}^{\top} \\ \Omega_{1,N-1}^{(t)} + \frac{1}{N}A_{Nt}X_{Nt}X_{Nt}^{\top} & \Sigma_{N-1}^{(t)} + \frac{1}{N}X_{Nt}X_{Nt}^{\top} \end{pmatrix},$$

and

$$H_t = \frac{1}{N}\sum_{i=1}^{N} \Psi(X_{it})Z_{it}^{\top} = \left(\Omega_{2,N-1}^{(t)} + \frac{1}{N}A_{Nt}\Psi(X_{Nt})X_{Nt}^{\top} \quad \Xi_{N-1}^{(t)} + \frac{1}{N}\Psi(X_{Nt})X_{Nt}^{\top}\right).$$

Moreover, $\mathbf{U}_t$ denotes a matrix whose columns form an orthonormal basis for the orthogonal complement of the column space $\mathbf{C}_t^{\pi} = \mathbb{E}_{X_t}^{\pi}\left[\frac{1}{N}\mathbf{\Psi}_t^{\top}\mathbf{X}_t\right]$. We note that the cumulative reward $\sum_{t=1}^{T} R_{N,t}$ coincides with the objective function to be optimized on the last day.

Finally, note that the reward $R_{N,t}$ depends only on $\mathcal{S}_{N-1,t}$ and the information observed on the $N$-th day up to time $t$. Accordingly, we define the state at time $t$ as

$$S_{N,t} = \{\mathcal{S}_{N-1,t}, \underbrace{X_{N,1}, A_{N,1}, \ldots, X_{N,t}, \varphi(t)}_{\mathcal{H}_{N,t}}\}. \tag{A.16}$$

Here, $\varphi(t)$ denotes the time-varying exogenous features.

The algorithm is summarized in Algorithm 4 and it is worth making the following additional clarifications.

- **Behavior-policy class $\pi_b$.** The past-summary feature $\mathcal{S}_{N-1,t}$ is constructed from the dataset collected over the previous $(N-1)$ days, $\{(X_{it}, A_{it}) : 1 \leq i \leq N-1, 1 \leq t \leq T\}$, under a behavior-policy class $\pi_b$. In principle, one would like $\pi_b$ to cover as many action-allocation policies as possible. However, enumerating all such policies is prohibitively large and unrealistic. Moreover, certain extreme treatment schemes are clearly undesirable in practice, as they can lead to ill-conditioned (or even non-invertible) matrices in the objective and hence numerical instability, similar to issues encountered in the single-stage setting. Therefore, we restrict $\pi_b$ to a collection of simple and commonly used policy classes, which include uniform random, switchback mechanisms and so on.

- **The RL state of $N$th day.** Our RL state representation incorporates historical information from the past $(N-1)$ days, the current-day information, and time-varying exogenous features. This enriched state specification increases the likelihood that the Markov property holds (i.e., that the problem can be well approximated by an MDP).

- **The RL reward**. For each $t$, our instantaneous RL reward depends on information from the previous $(N-1)$ days; thus, we incorporate the past-summary feature $\mathcal{S}_{N-1,t}$ in its computation.

---

**Algorithm 4** Policy learning on the final day via RL

---

1: **Input:** number of epochs $K$; number of rollouts $B$; behavior-policy class $\pi_b$; design priors $\{(\mathbf{u}_t, \sigma_e^2(t,t))\}_{t=1}^T$; initialized policy network $\pi_N$; transition model $\{\mathcal{P}_t(\cdot \mid x, a)\}_{t=2}^T$; initial distribution $\mathcal{P}_1(\cdot)$ for $X_{N,1}$.
2: **for** $k = 1, \ldots, K$ **do**
3:     Generate $B$ datasets for the past $N-1$ days under $\pi_b$: $\{(X_{it}^{(b)}, A_{it}^{(b)}) : 1 \leq i \leq N-1, 1 \leq t \leq T\}_{b=1}^B$.
4:     **for** $b = 1, \ldots, B$ **do**
5:         Sample $X_{N,1}^{(b)} \sim \mathcal{P}_1(\cdot)$ and set $S_{N,1}^{(b)}$ by (A.16).
6:         Sample action $A_{N,1}^{(b)} \sim \pi_N(\cdot \mid S_{N,1}^{(b)})$ and compute $R_{N,1}^{(b)}$ via (A.15).
7:         **for** $t = 2, \ldots, T$ **do**
8:             Sample $X_{N,t}^{(b)} \sim \mathcal{P}_t(\cdot \mid X_{N,t-1}^{(b)}, A_{N,t-1}^{(b)})$ and set $S_{N,t}^{(b)}$ by (A.16).
9:             Sample action $A_{N,t}^{(b)} \sim \pi_N(\cdot \mid S_{N,t}^{(b)})$ and compute $R_{N,t}^{(b)}$ from $(S_{N,t}^{(b)}, A_{N,t}^{(b)})$ via (A.15).
10:         **end for**
11:     **end for**
12:     Update the policy network $\pi_N$ using the collected batch data via the A2C method.
13: **end for**
14: **Output:** Trained policy $\pi_N^* \leftarrow \pi_N$.

---

**Algorithm for the $n$-th day.** We now consider the decision-making problem on the $n$-th day, which integrates RL within each day and DP across days. By this stage, the full history from the previous $n-1$ days has been observed. Suppose we aim to learn a policy $\pi_n$ for the $n$-th day. Importantly, the resulting MSE depends on the past only through a set of summary statistics,

$$\mathcal{S}_{n-1,t} = \{\Sigma_{n-1}^{(t)}, \Omega_{1,n-1}^{(t)}, \Xi_{n-1}^{(t)}, \Omega_{2,n-1}^{(t)}\}.$$

Consequently, decisions at time $t$ can only be based on this summarized past information together with the observations collected on the $n$-th day up to time $t$. We therefore define the state at time $t$ as

$$S_{n,t} = \{\mathcal{S}_{n-1,t}, \underbrace{X_{n,1}, A_{n,1}, \ldots, X_{n,t}, \varphi(t)}_{:=\mathcal{H}_{n,t}}\}. \tag{A.17}$$

Here, $\varphi(t)$ denotes the time-varying exogenous features.

However, to evaluate the contribution of the current action $A_{nt}$ at time $t$ to the overall objective, it is necessary to account for its impact on the future $N-n$ days. This is precisely where DP comes into play. Specifically, we fix the policies for future days at their optimal values $\{\pi_i^*\}_{i=n+1}^N$ and generate future data accordingly. Conditioned on both the realized trajectory under $\pi_n$ and the simulated trajectories induced by the future policies, we can compute the immediate reward

$$R_{n,t} = -\frac{1}{T}\left\{\frac{\sigma_t^2}{N}\mathbf{u}_t^\top G_t^{-1}\mathbf{u}_t + \eta_t^2\|\mathbf{U}_t^\top H_t G_t^{-1}\mathbf{u}_t\|_2^2\right\}. \tag{A.18}$$

Here,

$$G_t = \frac{1}{N}\sum_{i=1}^N Z_{it}Z_{it}^\top = \begin{pmatrix} \Sigma_{n-1}^{(t)} + \frac{1}{N}\sum_{i=n}^N X_{it}X_{it}^\top & \Omega_{1,n-1}^{(t)} + \frac{1}{N}\sum_{i=n}^N A_{it}X_{it}X_{it}^\top \\ \Omega_{1,n-1}^{(t)} + \frac{1}{N}\sum_{i=n}^N A_{it}X_{it}X_{it}^\top & \Sigma_{n-1}^{(t)} + \frac{1}{N}\sum_{i=n}^N X_{it}X_{it}^\top \end{pmatrix},$$

and

$$H_t = \frac{1}{N}\sum_{i=1}^N \Psi(X_{it})Z_{it}^\top = \begin{pmatrix} \Omega_{2,n-1}^{(t)} + \frac{1}{N}\sum_{i=n}^N A_{it}\Psi(X_{it})X_{it}^\top & \Xi_{n-1}^{(t)} + \frac{1}{N}\sum_{i=n}^N \Psi(X_{it})X_{it}^\top \end{pmatrix}.$$

Moreover, $\mathbf{U}_t$ denotes a matrix whose columns form an orthonormal basis for the orthogonal complement of the column space $\mathbf{C}_t^\pi = \mathbb{E}_{X_t}^\pi\left[\frac{1}{N}\mathbf{\Psi}_t^\top \mathbf{X}_t\right]$.

Finally, the cumulative reward $\sum_{t=1}^T R_{n,t}$ corresponds to the objective function to be optimized for the $n$-th day.

We summarize the overall procedure in Algorithm 5 and provide the following additional clarifications.

- When $n = 1$, we set $\mathcal{S}_{0,t} = \emptyset$, and the state reduces to $S_{1,t} = \{\mathcal{H}_{1,t}\}$ for all $t$.

- When training the policy $\pi_n$, the immediate reward in (A.18) depends on three components: (i) past information summarized by $\mathcal{S}_{n-1,t}$, (ii) data generated by the current rollout under $\pi_n$, and (iii) information induced by the future optimal policies $\{\pi_i^*\}_{i=n+1}^N$. This decomposition reflects the core DP principle.

---

**Algorithm 5** Policy learning for day $n$ via RL and DP

---

1: **Input:** number of epochs $K$; number of rollouts $B$; behavior-policy class $\pi_b$; design priors $\{(\mathbf{u}_t, \sigma_e^2(t,t))\}_{t=1}^T$; initialized policy network $\pi_n$; transition model $\{\mathcal{P}_t(\cdot \mid x, a)\}_{t=2}^T$; initial distribution $\mathcal{P}_1(\cdot)$ for $X_{n,1}$; learned future-day policies $\{\pi_i^*\}_{i=n+1}^N$.
2: **for** $k = 1, \ldots, K$ **do**
3:     Generate $B$ datasets for the past $n - 1$ days under $\pi_b$: $\{(X_{it}^{(b)}, A_{it}^{(b)}) : 1 \leq i \leq n - 1, 1 \leq t \leq T\}_{b=1}^B$.
4:     **for** $b = 1, \ldots, B$ **do**
5:         Sample $X_{n,1}^{(b)} \sim \mathcal{P}_1(\cdot)$ and set $S_{n,1}^{(b)}$ via (A.17).
6:         Sample action $A_{n,1}^{(b)} \sim \pi_n(\cdot \mid S_{n,1}^{(b)})$.
7:         **for** $t = 2, \ldots, T$ **do**
8:             Sample $X_{n,t}^{(b)} \sim \mathcal{P}_t(\cdot \mid X_{n,t-1}^{(b)}, A_{n,t-1}^{(b)})$ and set $S_{n,t}^{(b)}$ via (A.17).
9:             Sample action $A_{n,t}^{(b)} \sim \pi_n(\cdot \mid S_{n,t}^{(b)})$.
10:         **end for**
11:         *Generate future $(N - n)$ days' datasets $\{(X_{it}^{(b)}, A_{it}^{(b)}) : n + 1 \leq i \leq N, 1 \leq t \leq T\}$ using policies $\{\pi_i^*\}_{i=n+1}^N$.*
12:         Compute immediate rewards $\{R_{n,t}^{(b)}\}_{t=1}^T$ via (A.18).
13:     **end for**
14:     Update the policy network $\pi_n$ using the collected batch data via the A2C method.
15: **end for**
16: **Output:** Trained policy $\pi_n^* \leftarrow \pi_n$.

---

## C. Additional Experiments

In this section, we provide additional details for Section 5, including the DGP and additional experimental results.

### C.1. Contextual Bandits (Continued)

**DGPs.** We assume the covariates follow a multivariate distribution where $X_{i,1} \equiv 1$, and the continuous covariates $(X_{i,2}, X_{i,3})^\top$ are drawn i.i.d. from $\mathcal{N}_2(\mathbf{0}, \Sigma)$ with a covariance matrix whose diagonal entries are equal to one and whose off-diagonal entries are 0.3. The noise terms $e_i$ are drawn from $\mathcal{N}(0, \sigma^2)$ with $\sigma^2 = 1$. The treatment is $A_i \in \{\pm 1\}$.

We consider two setups for the outcome $Y_i$:

**Setup 1:** $Y_i = A_i + \mu(X_i) + e_i$, where the baseline mean function is defined as:

$$\mu(X_i) = 1.2 - 1.4X_{i,2} + 0.8X_{i,3} + \sum_{j=2}^3 \left[ 2\sin(X_{i,j} + 0.5) + 1.5\cos(X_{i,j}^2 - 0.5) \right].$$

**Setup 2:** $Y_i = A_i + 0.6X_{i,2}A_i - 0.5X_{i,3}A_i + \mu(X_i) + e_i$.

We utilize a Legendre polynomial basis $[-1, 1]$: $\Psi(x) = \left( P_0(x_1), P_1(x_1), P_2(x_1), P_3(x_1), P_1(x_2), P_2(x_2), P_3(x_3) \right)^\top$.

**Experimental details.** In the single-stage experiments, we set the sample size $M = 5000$, the batch size $B = 8000$, and the regularization parameter $\nu^2 = 0.005$.

For the DNN, we employ a fully connected MLP architecture optimized for this task. The network consists of an input layer followed by four hidden layers with 512, 256, 128, and 64 units, respectively. Each hidden layer incorporates the following

components in order: Linear Transformation; Batch Normalization; Swish Activation; Dropout with rates set to 0.1, 0.1, 0.08, and 0.05 for the respective layers. The final output layer is a single linear unit.

Furthermore, we assume access to a large historical covariate dataset, allowing us to approximate the population moments $(\Sigma, \Xi, \mathbf{U})$ using their empirical counterparts.

### C.2. Dynamic Settings (Continued)

In this section, we provide detailed specifications for the experiments conducted in Section 5.2. For each day $i$, the initial state $X_{i,2:4}$ is sampled from a standard trivariate normal distribution. At each time step $t$, a binary action $A_{i,t}$ is allocated. We explicitly note that the state $X_{i,t,2:4}$ is defined as a continuous random variable in $\mathbb{R}^3$, whereas the action $A_{i,t}$ takes values in the binary set $\{-1, 1\}$. Subsequently, the system evolves according to the outcome and state transition models described below. The resulting data is aggregated into a dataset $\mathcal{D} = \{(X_{i,t}, A_{i,t}, Y_{i,t})\}_{i=1,t=1}^{N,T}$. In our experiments, we consider time horizons $T \in \{6, 12\}$ and vary the sample size $N \in \{21, 28, 35, 42\}$.

Specifically, the outcome $Y_{i,t}$ is generated according to:

$$Y_{i,t} = b_t + \boldsymbol{\beta}_t^\top X_{i,t,2:4} + \gamma_t A_{i,t} + A_{i,t}\mathbf{c}_t^\top X_{i,t,2:4} + \delta \sum_{k=2}^{4} \cos(X_{i,t,k}) + \epsilon_{i,t}, \quad t = 1, \ldots, T. \tag{A.19}$$

In this formulation, $b_t$ represents the intercept, while $\gamma_t$ and $\boldsymbol{\beta}_t \in \mathbb{R}^3$ denote the main effects of the action $A_{i,t}$ and the covariate vector $X_{i,t}$, respectively. The vector $\mathbf{c}_t \in \mathbb{R}^3$ captures the linear interaction effect between the action and the covariate. To assess robustness against model misspecification, we include a nonlinear term weighted by $\delta$, where $X_{i,t,k}$ is the $k$-th component of the time-varying covariate vector. The parameter $\delta$ controls the intensity of this nonlinearity, and we vary $\delta \in \{1, 2, 3\}$ to correspond to low, medium, and high levels of misspecification. Finally, the noise terms $\epsilon_{i,t}$ are independent and identically distributed (i.i.d.) across all samples and time steps, following a standard normal distribution $\mathcal{N}(0, 1)$.

The state transition dynamics from $t = 1$ to $T - 1$ are governed by:

$$X_{i,t+1,2:4} = \boldsymbol{\phi}_{0,t} + \boldsymbol{\Phi}_t X_{i,t,2:4} + \boldsymbol{\alpha}_t A_{i,t} + A_{i,t}\boldsymbol{\Xi}_t X_{i,t,2:4} + \mathbf{E}_{i,t}. \tag{A.20}$$

The term $\boldsymbol{\phi}_{0,t}$ represents the intercept, while $\boldsymbol{\Phi}_t \in \mathbb{R}^{3\times3}$ defines the baseline autoregressive dynamics. The influence of the action is captured by two terms: $\boldsymbol{\alpha}_t \in \mathbb{R}^3$ represents the direct main effect, and the matrix $\boldsymbol{\Xi}_t \in \mathbb{R}^{3\times3}$ models the interaction between the state and action, allowing the action to modulate the transition mechanism. Finally, $\mathbf{E}_{i,t}$ represents the transition noise, which is i.i.d. and follows a multivariate normal distribution $\mathcal{N}(\mathbf{0}, 1.5\mathbf{I}_3)$.

**Parameters.** We outline how the parameters for the outcome and transition dynamics are generated. To ensure parameter diversity, the scalar intercept $b_t$ and each entry of the intercept vector $\boldsymbol{\phi}_{0,t}$ are sampled from the uniform distribution over the disjoint union Uniform($[-1.0, -0.5] \cup [0.5, 1.0]$). For the outcome model, the elements of the state coefficient vector $\boldsymbol{\beta}_t$ and the interaction vector $\mathbf{c}_t$ are sampled element-wise from Uniform($[-0.3, -0.1] \cup [0.1, 0.3]$) and Uniform($[-0.1, -0.05] \cup [0.05, 0.1]$), respectively. Regarding the action effects, $\gamma_t$ is drawn from Uniform$[0.5, 0.8]$, while the components of $\boldsymbol{\alpha}_t$ are i.i.d. variables following $\mathcal{N}(0, 0.09)$. Finally, the entries of the transition matrices $\boldsymbol{\Phi}_t$ and $\boldsymbol{\Xi}_t$ are drawn from Uniform$[-0.3, 0.3]$ and Uniform$[-0.2, 0.2]$, respectively.

**Additional Results: Nonlinearity Degree $\delta$ and Time Horizon $T$.** In Section 5.2, we evaluated our method under strong nonlinearity ($\delta = 3$). Here, we further report results for $\delta = 1$ and $\delta = 2$ to assess performance across varying levels of nonlinearity and time horizons $T$.

As shown in Figure 4, the empirical MSE of all methods increases markedly as the nonlinearity becomes stronger (from small to moderate bias), reflecting the fact that the underlying ATE estimators are built on OLS-style linear modeling and thus become more vulnerable under nonlinear misspecification. Despite this degradation, our method remains the most stable and consistently achieves the lowest MSE across all horizons ($T = 6, 12$) and experiment lengths ($n = 21, 28, 35, 42$). This robustness stems from explicitly incorporating and controlling the bias induced by the nonlinear component, whereas competing designs do not directly guard against such misspecification.

Moreover, since the data-generating process is inherently an MDP, **TMDP** provides a strong baseline and performs competitively in several regimes. However, approaches that rely on (approximately) memoryless/randomized allocations

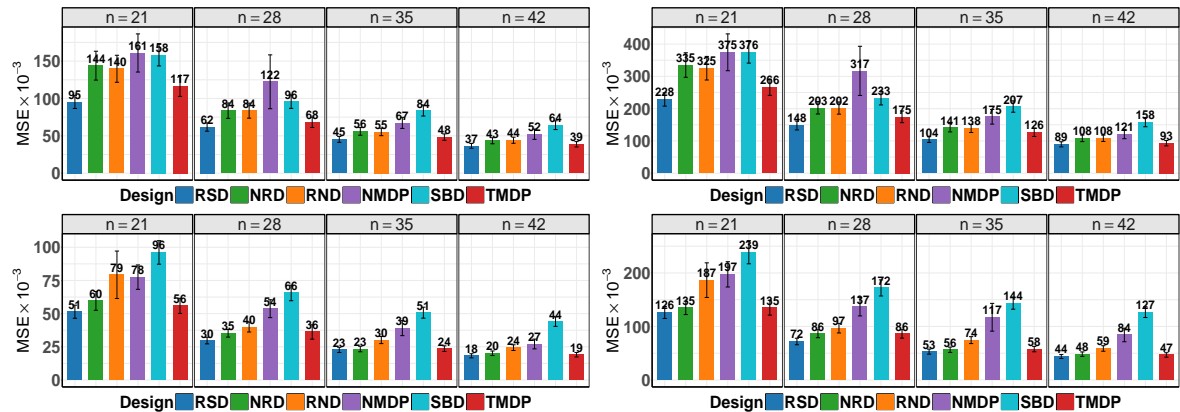

*Figure 4.* Empirical MSE (95% CI) under the dynamic settings: with small bias, $T = 6$ (top left), with moderate bias, $T = 6$ (top right); with small bias, $T = 12$ (bottom left), with moderate bias, $T = 12$ (bottom right).

do not fully exploit the sequential structure of the experiment to improve future assignments. In contrast, our sequential allocation strategy leverages accumulated information from past allocations and outcomes to optimize the remaining trajectory, yielding uniformly better performance than **TMDP**.

**Experimental details.** In the case of $T = 6$, we set $B = 20,000$, $K = 300$, and $\eta_t = 1.0$. For the actor network, we use a two-layer MLP with GELU activations, where each hidden layer has 256 units. A linear projection is applied to map the final hidden representation to a scalar logit. For the critic network, we share the same first two layers as the actor network, and then add an additional linear layer followed by a GELU activation. Finally, we project to a scalar and apply a Softplus activation function.

## C.3. Real-Data-Based Simulation (Continued)

The A/A experiment spans $D = 40$ days, during which a single order dispatch policy was employed throughout. While the trajectory data was originally recorded in 24 non-overlapping intervals per day, we aggregate these into a single observation per day to align with our i.i.d. setting. Including the intercept, the resulting covariate vector $X_d$ is three-dimensional, consisting of an intercept and two marketplace features: (i) the total number of order requests and (ii) the total driver online time during the previous day. These features capture the demand and supply dynamics of the two-sided marketplace, respectively. The outcome $Y_d$ is defined as the total driver income earned within the day.

**Experimental details.** We first fit a linear model to the A/A data $\{Y_d, X_d\}_{d=1}^D$ using OLS to obtain the estimated coefficient $\widehat{\beta}$, the residuals $\{\widehat{e}_d\}_{d=1}^D$, and the error variance $\widehat{\sigma}^2$. Additionally, we compute the empirical moments of the covariates $(\Sigma, \Xi, \mathbf{U})$ based on $\{X_d\}_{d=1}^D$ to approximate the population moments. We specify the treatment effect parameters as $\widehat{\gamma} = 0.025 \times \bar{Y}$ and $\widehat{\xi} = 0.0125 \times \bar{X}$, where $\bar{Y}$ and $\bar{X}$ denote the sample means.

To evaluate the performance of different designs, we generate synthetic data using a parametric bootstrap approach. Specifically, for each unit $i = 1, \ldots, N$, the outcomes are generated as follows:

$$\text{Without Interaction: } \widetilde{Y}_i = \widehat{\gamma} A_i + \widehat{\beta}^\top X_i + \mu'(X_i) + \varepsilon_i,$$
$$\text{With Interaction: } \widetilde{Y}_i = A_i X_i^\top \widehat{\xi} + \widehat{\beta}^\top X_i + \mu'(X_i) + \varepsilon_i,$$

where $\varepsilon_i \overset{i.i.d.}{\sim} \mathcal{N}(0, \widehat{\sigma}^2)$ and the covariates $X_i$ are resampled from the empirical distribution. To introduce stronger non-linearity beyond the fitted linear components, we incorporate a baseline mean function $\mu'(X_i)$, defined as:

$$\mu'(X_i) = 1.2 - 1.4X_{i,2} + 0.8X_{i,3} + \sum_{j=2}^3 \left[2\sin(X_{i,j} + 0.5) + 2\cos(X_{i,j}^2 - 0.5)\right].$$

This setup allows us to accurately estimate the ground truth ATE via Monte Carlo simulation.

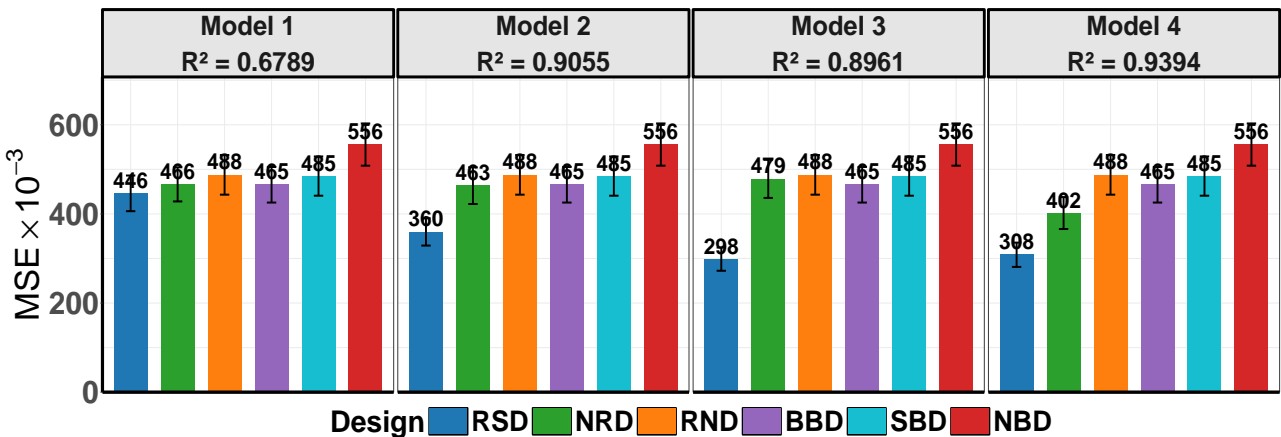

*Figure 5.* Empirical MSE with 95% confidence intervals under the contextual bandit setting with additive treatment effects.

Furthermore, we set $M = 12000$, $B = 10000$, and $\nu^2 = 0.005$. For the DNN, we employ a custom-designed 7-layer MLP. The network architecture consists of an input layer followed by six hidden layers with decreasing widths of 512, 256, 128, 64, 32, and 16 units, respectively, and a final linear output layer. Each hidden layer follows a standardized block structure: Linear Transformation; Batch Normalization; Swish Activation; Dropout, with rates progressively set to 0.2, 0.15, 0.1, 0.1, 0.05, and 0.05 for the six hidden layers to prevent overfitting.

### C.4. Sensitivity Analysis to Value Function Estimation Errors

In this section, we assess the sensitivity of the proposed method to value function approximation errors under the experimental setting in Section 5.1. Specifically, we train four fully connected neural networks with architectures

$$(128, 64, 32), \quad (256, 128, 64, 32), \quad (256, 128, 64, 16), \quad \text{and} \quad (512, 256, 128, 64, 32).$$

All networks are trained under a common pipeline with batch normalization, SiLU activation, and dropout. These models achieve $R^2$ values of 0.6789, 0.9055, 0.8961, and 0.9394, respectively, against the ground-truth value function on the last day.

We then evaluate the downstream MSE of the ATE estimator. As shown in Figure 5, our method, as well as NRD, remains stable across different network architectures. Notably, even when the value function is estimated with only moderate accuracy, e.g., $R^2 = 0.6789$, our method still outperforms the baseline designs. This suggests that the proposed design is relatively robust to approximation errors in value function estimation.

### C.5. Efficiency Analysis and Tightness of the Proposed Upper Bound

In this section, we first empirically examine the efficiency loss of the proposed design under correct model specification, where the bias term vanishes and the MSE reduces to the variance term. Second, we discuss a condition under which the proposed upper bound is attainable, showing that the bound is not merely a loose analytical artifact.

(1) Under correct model specification, a natural lower benchmark is given by the design that directly minimizes the true MSE. In this case, the misspecification bias is zero, and the remaining discrepancy between our robust upper bound and the true MSE reflects the worst-case bias component introduced to guard against model misspecification. To quantify the potential price paid for such robustness, we follow the robust design literature (Wiens, 2000) and evaluate the relative efficiency of different designs under $f(X) = 0$.

In our setting, when $f(X) = 0$, the sequential design of Bhat et al. (2020), denoted by **NRD**, is optimal in the sense that it minimizes the true MSE. We therefore use **NRD** as the oracle benchmark under correct specification and define the efficiency of a candidate design $d$ as

$$\text{Eff}(d) := \frac{\text{MSE}(\textbf{NRD})}{\text{MSE}(d)}.$$

By construction, $\text{Eff}(d) \le 1$, and values closer to one indicate that the design $d$ incurs only a small efficiency loss relative to **NRD** under correct model specification.

Table 1 reports this efficiency measure for the proposed design **RSD**, as well as for the baseline designs **BBD**, **SBD**, and **NBD**, under the experimental setup in Section 5.1 with correct model specification. Table 2 reports the corresponding MSE values. As expected, **NRD** achieves the smallest MSE under correct specification. Nevertheless, **RSD** performs comparably to **NRD**, achieving efficiency levels of approximately 97%–99% across all sample sizes and consistently outperforming the other baseline designs. This suggests that the robustness improvement obtained by **RSD** comes at only a small efficiency cost when the model is correctly specified.

(2) We next discuss the attainability of the proposed upper bound. The bound is derived by applying the Cauchy–Schwarz inequality to the misspecification-induced bias term. Therefore, the bound becomes tight when the misspecification direction aligns with the worst-case direction identified by the design criterion. More specifically, when the misspecification function $f$ can be represented as

$$f(\mathbf{X}) = \eta \mathbf{\Psi}(\mathbf{X}) \mathbf{U} \boldsymbol{\kappa},$$

where $\boldsymbol{\kappa}$ is proportional to $\mathbf{a}^\top \mathbf{\Psi}(\mathbf{X}) \mathbf{U}$ under a unit $\ell_2$-norm constraint, the Cauchy–Schwarz inequality is attained. In this case, the proposed upper bound is asymptotically equivalent to the true MSE; see the proof of Proposition A.1 and Equation (A.3). This provides a theoretical justification that the proposed upper bound can be achieved under aligned misspecification directions, while the empirical results above show that optimizing the bound incurs little efficiency loss under correct specification.

*Table 1.* Median relative efficiency with respect to the baseline design **NRD** under the contextual bandit setting with additive treatment effects and $f(X) = 0$. Larger values indicate better efficiency relative to **NRD**.

| Design | $N = 21$ | $N = 28$ | $N = 35$ | $N = 42$ |
|--------|----------|----------|----------|----------|
| RSD | 0.9910 | 0.9855 | 0.9778 | 0.9886 |
| RND | 0.9144 | 0.9321 | 0.9391 | 0.9486 |
| BBD | 0.9721 | 0.9708 | 0.9725 | 0.9737 |
| SBD | 0.9691 | 0.9657 | 0.9683 | 0.9704 |
| NBD | 0.9379 | 0.9542 | 0.9556 | 0.9606 |

*Table 2.* MSE of the ATE estimator under the contextual bandit setting with additive treatment effects and $f(X) = 0$. Lower values indicate better estimation accuracy.

| Design | $N = 21$ | $N = 28$ | $N = 35$ | $N = 42$ |
|--------|----------|----------|----------|----------|
| RSD | 0.2135 | 0.1491 | 0.1174 | 0.1027 |
| NRD | 0.1958 | 0.1493 | 0.1124 | 0.0934 |
| RND | 0.2284 | 0.1554 | 0.1307 | 0.1052 |
| BBD | 0.2111 | 0.1512 | 0.1198 | 0.0991 |
| SBD | 0.2018 | 0.1511 | 0.1175 | 0.1107 |
| NBD | 0.2380 | 0.1552 | 0.1193 | 0.1084 |

## D. Additional Discussions on General Settings

In this section, we discuss two possible extensions of the proposed framework. The first concerns heteroskedastic errors, where the noise variance may vary across units. The second concerns a more general misspecification structure in which the misspecification term may depend on the assigned treatment.

### D.1. Discussion on Heteroskedastic Errors

Our proposed framework can be extended to heteroskedastic settings, although the resulting optimization problem becomes more challenging. Following the robust design perspective of Wiens (2000), suppose that

$$\text{Var}(e_i \mid A_i, X_i) = \sigma^2 g_i,$$

where the heteroskedasticity factors $g_i > 0$ are unknown. To account for nonconstant error variance, we replace the ordinary least squares estimator with a weighted least squares estimator using

$$\mathbf{W} = \text{diag}(w_1, \ldots, w_N).$$

The resulting MSE decomposition has a structure similar to that in Eq. (3). The main difference is that the variance term now depends on the unknown factors $g_i$, while the bias term induced by the misspecification function $f$ can be handled analogously to the homoskedastic case by working with the transformed design matrix $\mathbf{W}^{1/2}\mathbf{Z}$.

To obtain a robust criterion against unknown heteroskedasticity, we consider the worst-case MSE over

$$\sum_{i=1}^{N} g_i^2 \leq N.$$

The variance component can be written as

$$I_1^W = \sigma^2 \sum_{i=1}^{N} g_i w_i^2 \left\{ \mathbf{u}^\top (\mathbf{Z}^\top \mathbf{W} \mathbf{Z})^{-1} Z_i \right\}^2.$$

By the Cauchy–Schwarz inequality, this term is bounded by

$$I_1^W \leq \sigma^2 \sqrt{N} \left[ \sum_{i=1}^{N} w_i^4 \left\{ \mathbf{u}^\top (\mathbf{Z}^\top \mathbf{W} \mathbf{Z})^{-1} Z_i \right\}^4 \right]^{1/2}.$$

This bound provides a natural robust counterpart to the homoskedastic variance term.

However, unlike in the homoskedastic case, the resulting objective involves fourth-order terms in the assignment-dependent quantities. In particular, the term

$$\mathbf{u}^\top (\mathbf{Z}^\top \mathbf{W} \mathbf{Z})^{-1} Z_i$$

depends on the treatment assignments through both $A_i$ and aggregate weighted assignment summaries such as $\sum_{i=1}^{N} w_i A_i$. After substitution into the bound above, the objective therefore contains higher-order interactions among assignments, making the optimization substantially more complex than the objective studied in the main paper. Whether the dynamic programming strategy developed for the homoskedastic setting can be efficiently adapted to this heteroskedastic objective is nontrivial. We therefore leave the development of efficient algorithms for this extension to future work.

### D.2. Discussion on Treatment-Dependent Misspecification

In the main paper, we focus on misspecification of the covariate effect through a common function $f(X)$ for simplicity. Nevertheless, the proposed framework may be extended to a more general setting where the misspecification depends on the assigned treatment.

Specifically, for each $a \in \{1, -1\}$, consider the treatment-specific conditional mean model

$$m_a(X) = X^\top \theta^{(a)} + f^{(a)}(X),$$

where

$$m_a(X) = \mathbb{E}(Y \mid A = a, X),$$

and $\theta^{(a)}$ is defined as the best linear approximation of $m_a(X)$ onto the span of $X$. Since the first component of $X$ is an intercept, the corresponding first-order condition implies

$$\mathbb{E}\{f^{(a)}(X)\} = 0, \qquad a \in \{1, -1\}.$$

It follows that the average treatment effect can be expressed as

$$\text{ATE} = \mathbb{E}\{m_1(X) - m_{-1}(X)\} = \mathbf{u}^\top \left( \theta^{(1)} - \theta^{(-1)} \right),$$

where $\mathbf{u} = \mathbb{E}(X)$.

A natural estimator can then be constructed by separately estimating $\theta^{(1)}$ and $\theta^{(-1)}$ using the treatment and control groups, respectively, and forming the plug-in estimator

$$\widehat{\text{ATE}} = \mathbf{u}^\top \left( \widehat{\theta}^{(1)} - \widehat{\theta}^{(-1)} \right).$$

The conditional MSE of this estimator admits a decomposition analogous to that in Appendix A.1, with variance terms arising from the two treatment-specific regressions and bias terms arising from $f^{(1)}$ and $f^{(-1)}$. Using a similar orthogonalization argument, one can derive a corresponding upper bound for the MSE and formulate a robust design objective that guards against treatment-specific misspecification.

The full sequential implementation of this extension, however, requires additional technical development. In particular, the design would need to track treatment-specific summary statistics, such as separate Gram matrices and imbalance summaries for the two treatment arms, rather than the single set of summaries used in the main paper. This enlarges the state space of the sequential optimization problem and may require new computational strategies. We therefore view treatment-dependent misspecification as a natural extension of our framework, but leave a complete sequential design and optimization procedure for future work.

