# OpenReview forum: "Robust Sequential Experimental Design for A/B Testing"
_ICML.cc/2026/Conference — ICML 2026 regular_

### Official Review · Reviewer_GhJw · 2026-02-19

**Soundness:** 3
**Presentation:** 3
**Significance:** 3
**Originality:** 3
**Overall Recommendation:** 5
**Confidence:** 2

**Summary:**

This work studies robust sequential experimental design for A/B testing in the contextual and dynamic settings. In the first setting, the outcome is assumed to consist of a fixed treatment effect, a linear term about the covariates, a general, non-parametric term about the covariates, and an i.i.d. random error. The goal is to minimize (an upper bound) of the MSE of the estimated ATE. The authors propose a dynamic programming method that selects the next action (control or treatment) based on current information. The value function in the DP method is trained via DNN.

**Compliance With Llm Reviewing Policy:**

Affirmed.

**Key Questions For Authors:**

1. On top right of page 4, could you explain why the approximation error f must satisfy the orthogonality condition?

2. In Algorithm 1, what is \pi_b and how to select/determine it?

**Limitations:**

Yes.

**Strengths And Weaknesses:**

Good.

---

> ### Author Rebuttal · Authors · 2026-03-31
>
> We sincerely thank Reviewer GhJw for the appreciation of our work and for providing constructive comments.
>
> - **Q1: On page 4, explain why $f$ must satisfy the orthogonality condition.**
>
>   We thank the reviewer for this question. This orthogonality condition follows directly from the definition of $\theta$, which minimizes the mean squared error between the outcome regression model $m(A, X)$ and its linear approximation. Taking the derivative of the objective function, $\theta$ must satisfy the following first-order condition: $\mathbb{E}\left[X\{m(A,X)-(A,X^\top)\theta\}\right]=0$. Under our additive model assumption, $m(A,X)=(A,X^\top)\theta+f(X)$. Substituting the residual term $m(A,X)-(A,X^\top)\theta=f(X)$ into the first-order condition yields the orthogonality condition $\mathbb{E}[Xf(X)]=0$. We will clarify this point if accepted.
>
> - **Q2: In Algorithm 1, what is $\pi_b$ and how is it selected?**
>
>   The behavior policy $\pi_b$ in Algorithm 1 is used to generate rollout trajectories for estimating the value function $Q_n$. Specifically, $Q_n$ depends on the imbalance statistics $(\Delta_n, \Gamma_n)$. Ideally, one would enumerate all possible values of $(\Delta_n, \Gamma_n)$, but this is computationally infeasible. Instead, we use a behavior policy $\pi_b$ to explore the space of imbalance statistics. The goal of $\pi_b$ is to generate a sufficiently diverse set of $(\Delta_n, \Gamma_n)$ values in order to improve the accuracy of value function estimation.
>
>   In our implementation, for each $k = 0, 1, \dots, n$, we construct a canonical action sequence of length $n$ containing exactly $k$ entries equal to $+1$ and $n-k$ entries equal to $-1$. This yields $n+1$ action sequences with different numbers of treated and control subjects. These sequences define the behavior policy $\pi_b$. Our intuition is that the distribution of the imbalance statistics $(\Delta_n, \Gamma_n)$ depends critically on the difference between the numbers of treated and control subjects. This motivates our choice of $\pi_b$, which systematically spans all possible levels of imbalance.

---

### Official Review · Reviewer_Km6o · 2026-03-10

**Soundness:** 3
**Presentation:** 4
**Significance:** 2
**Originality:** 3
**Overall Recommendation:** 4
**Confidence:** 2

**Summary:**

This paper studies the setting of estimating ATE in a sequential A/B test using a linear model with some non-linear model misspecification in the data generation process. Their contribution is three-fold. First, using an orthogonalization technique, they are able to find a robust upper bound on the MSE. Second, they are able to use a DP algorithm to design the A/B test to minimize the MSE upper bound in the contextual bandit setup. Lastly, they extend their results to the dynamic setting where prior assignments exert carryover effects on future outcomes.

**Compliance With Llm Reviewing Policy:**

Affirmed.

**Final Justification:**

My questions have been adequately addressed and I am maintaining my positive score.

**Key Questions For Authors:**

I would like to understand how the results would change if the misspecification in the DGP were a function of the treatment assigned.

**Limitations:**

Yes

**Strengths And Weaknesses:**

## Strengths and Weaknesses

### Soundness
The submission is technically sound and the claims are well-supported by the theoretical results.

### Presentation
The paper is written clearly and is well-structured.

### Significance
The paper addresses a relevant problem; studying A/B testing from a misspecification point of view is an important step toward making it more realistic. That said, one consideration worth exploring might be the inclusion of a lower bound analysis on the MSE, which would help clarify how tight the proposed upper bound is, and in turn, better justify the meaningfulness of optimizing for it.

### Originality
I find the problem setting to be original, though the technical analysis appears somewhat standard. I would kindly defer to the AE and the other reviewers, who are better positioned than I am to assess the degree of technical novelty.

---

> ### Author Rebuttal · Authors · 2026-03-31
>
> We thank the reviewer for the valuable comments.
>
> - **Significance: It may be useful to include an MSE lower bound to clarify the tightness of the proposed upper bound.**
>
>   This is an excellent suggestion. We address tightness from two perspectives.
>
>   First, a simple lower bound of the MSE arises under correct model specification. In this case, the bias term vanishes and the MSE reduces to the variance term alone. The gap between this lower bound and our upper bound can therefore be interpreted as the worst-case bias induced by misspecification. To quantify this gap numerically, we follow the robust design literature [1] and define an efficiency measure as the ratio of the MSE under our robust design to that under the design optimized for the correctly specified model, both evaluated at $f(X)=0$.
>
>   In our setup, when $f(X)=0$, the sequential design in [2], denoted by **NRD**, is optimal in the sense that it minimizes the true MSE. We therefore use **NRD** as the benchmark and measure the efficiency of a candidate design $d$ as
>
>   $$
>   \mathrm{Eff}(d) := \frac{\mathrm{MSE}(\text{NRD})}{\mathrm{MSE}(d)}.
>   $$
>
>   Values of Eff closer to 1 indicate that the design $d$ is nearly as efficient as **NRD** under correct model specification.
>
>   Table 1 reports this efficiency measure for the proposed design **RSD** as well as other baseline designs **BBD**, **SBD** and **NBD** under the setup in Section 5.1 with correct model specification, whereas [Table 2](https://www.dropbox.com/scl/fi/ccmtls97nctb1fkef9d0u/tab2_.png?rlkey=fcma6k8ap7yfuob4wrsdsjn5n&st=hkkl635h&dl=0) reports the corresponding MSE values.
>
>   | Design | $N=21$ | $N=28$ | $N=35$ | $N=42$ |
>   |--------|-------:|-------:|-------:|-------:|
>   | RSD    | 0.9910 | 0.9855 | 0.9778 | 0.9886 |
>   | RND    | 0.9144 | 0.9321 | 0.9391 | 0.9486 |
>   | BBD    | 0.9721 | 0.9708 | 0.9725 | 0.9737 |
>   | SBD    | 0.9691 | 0.9657 | 0.9683 | 0.9704 |
>   | NBD    | 0.9379 | 0.9542 | 0.9556 | 0.9606 |
>
>   **Table 1.** Median relative efficiency with respect to the baseline design **NRD** under the contextual bandit setting with additive treatment effects and $f(X)=0$. Larger values indicate better efficiency relative to **NRD**.
>
>   As expected, **NRD** achieves the smallest MSE under correct specification. However, the proposed design performs closely to **NRD**, with efficiency levels around 97% -- 99% across all sample sizes, consistently higher than those of the baseline designs. This indicates that the gain in robustness comes at only a small efficiency loss.
>
>   Second, our upper bound is sharp in a worst-case, design-dependent sense. Specifically, for a given treatment vector $\mathbf{a}$, if the misspecification function takes the form $f(X)=\eta \boldsymbol{\Psi}(\mathbf{X})\mathbf{U}\boldsymbol{\kappa}$ (see the proof of Proposition A.1), where $\boldsymbol{\kappa}^\top$ is proportional to $\mathbf{a}^\top\boldsymbol{\Psi}(\mathbf{X})\mathbf{U}$ and satisfies $\lVert \boldsymbol{\kappa} \rVert_2 = 1$, then the proposed upper bound is asymptotically equivalent to the true MSE. This follows from the Cauchy–Schwarz inequality used in deriving Eq. (A.3).
>
> - **Q1: how the results would change if the misspecification were a function of the treatment assigned?**
>
>   This is an excellent question. In the main paper, we focus on misspecification of the covariate effect $f(X)$, for simplicity. Nonetheless, our framework can be extended to the more general case where the misspecification depends on the treatment assignment. Specifically, for each $a \in $ {1,-1}, we consider the treatment-specific working model:
>
>   $$
>   m_{a}(X)=X^\top\theta^{(a)}+f^{(a)}(X),
>   $$
>
>   where $m_{a}(X)=\mathbb{E}(Y\mid A=a, X)$ and $\theta^{(a)}$ is defined as the best linear approximation of $m_{a}(X)$ onto the span of $X$. Since the first component of $X$ is an intercept, the first-order condition implies $\mathbb{E}[f_{a}(X)]=0$ for each $a$. Hence the ATE equals
>   $\text{ATE}=\mathbb{E}[m_{1}(X)-m_{-1}(X)]=\mathbf{u}^\top(\theta^{(1)}-\theta^{(-1)})$, where $\mathbf{u}=\mathbb{E}(X)$.
>
>   To estimate the ATE, one can separately estimate $\theta^{(1)}$ and $\theta^{(-1)}$ via OLS using data from the treatment and control groups, respectively, and then construct the corresponding plug-in ATE estimator. The resulting conditional MSE admits a decomposition analogous to that in Appendix A.1. Using a similar orthogonalization argument, one can derive a corresponding upper bound for the MSE and formulate an optimization objective. Extending the full sequential design requires additional technical development, such as tracking treatment-specific summary statistics, and is therefore beyond the scope of the current paper. We will add a discussion of this extension using the extra page if accepted.
>
> [1] Integer-valued, minimax robust designs for estimation and extrapolation in heteroscedastic, approximately linear models.
> [2] Near-optimal ab testing.

---

> > ### Author Rebuttal · Reviewer_Km6o · 2026-04-04
> >
> > Thank you to the authors for their response. I am maintaining my positive score.

---

> > > ### Author Response · Authors · 2026-04-04
> > >
> > > Thank you for your positive feedback. We sincerely appreciate your time and your continued positive evaluation.

---

### Official Review · Reviewer_j6i9 · 2026-03-12

**Soundness:** 4
**Presentation:** 4
**Significance:** 2
**Originality:** 2
**Overall Recommendation:** 4
**Confidence:** 4

**Summary:**

This paper studies robust experimental design under model misspecification, with application to contextual bandit and dynamic settings. They first establish an upper bound of MSE using an orthogonalization technique, and then optimize it using dynamic programming. They further extend their results to dynamic settings by integrating across-day DP and in-day RL. Experiments are conducted on both synthetic and real-world datasets.

**Compliance With Llm Reviewing Policy:**

Affirmed.

**Final Justification:**

I keep my positive scores as my concerns are resolved.

**Key Questions For Authors:**

- The theoretical results hugely depend on the correct sieve representation and the corresponding basis functions. How do you handle model misspecification due to incorrect sieve representation, especially in real-world problems where the representation space is unclear?
- The estimated value function is of critical importance in Algorithm 2. How does its estimation error affect the performance of the optimal treatment output? How do the authors ensure the behavior policy $\pi_b$ drives the state $X$ to sufficiently explore the representation space? I would appreciate it if the authors could justify the use of neural networks, either through error analysis or through methodologies that guarantee estimation quality.
- In Section 4, the authors still assume independence across the days. How restrictive are these assumptions, and how do they affect the practical performance in real-world problems?

**Limitations:**

yes

**Strengths And Weaknesses:**

**Strengths**: This paper is in-general well-written and smooth to follow, with a clear idea of optimizing the MSE upperbound to achieve robust estimation under model misspecification. The proofs of the theoretical results seem rigorous. The experiments are extensive and convincing.

**Weaknesses**: The originality of the paper is relatively weak. Currently, it's more likely to build directly on existing robust and DP-based sequential experimental design methods, and hence new conceptual breakthroughs are limited. Specifically:

- As the authors have pointed out, Bhat et al. (2020) are pioneers in formulating the sequential experimental design problem as a finite-time MDP and solving it with DP. They adopt the same MSE objective and use imbalance statistics as the states. The main additional component is an approximation error term $f(X_i)$ that makes the estimation of the value function hard, and the authors leverage neural networks to estimate it. It would be helpful if the authors could further clarify and emphasize how their method is substantively distinct from Bhat et al. (2020).

- Second, [1] also applies DP to sequential experimental design with function approximation for the value function. Moreover, they also go beyond linear outcome, and assume a general observation model accounting for modeling error. I believe the authors should cite this paper and discuss more explicitly how their approach differs from it.

*[1] Huan, X., & Marzouk, Y. M. (2016). Sequential Bayesian optimal experimental design via approximate dynamic programming. arXiv preprint arXiv:1604.08320.*

---

> ### Author Rebuttal · Authors · 2026-03-31
>
> Thank you for the valuable comments.
>
> * **W1: Comparison against [1]**
>  Our proposal differs from [1] in three main respects.
>
> 1. [1] optimizes exact MSE under a correctly specified linear model with additive effects and Gaussian covariates, while we allow misspecification via an unknown nonlinear $f(X)$ and optimize a worst-case upper bound via orthogonalization.
>
> 2. This leads to different algorithms: [1] exploits low-dimensional structure, whereas in our setting the worst-case upper bound renders exact DP intractable, so we use simulation-based rollouts and neural networks.
>
> 3. Our framework is more general, allowing interactive treatment effects and dynamic settings with carryover effects via a hierarchical DP/RL approach beyond [1].
>
> * **W2: Comparison against [2]**
>
>     Thank you for pointing out this paper. We will cite it if accepted. [2] and our work differ in several respects.
>
> 1. [2] studies Bayesian design for parameter inference using information gain, whereas we focus on ATE estimation in A/B testing and optimize a worst-case MSE upper bound from a frequentist perspective.
>
> 2. [2] assumes correct model specification, while we adopt a potentially misspecified model and ensure robustness to misspecification.
>
> 3. The DP formulations differ: [2] involves high-dimensional states requiring approximate DP with exploration–exploitation, whereas ours uses imbalance statistics, yielding a tractable recursion with function approximation via supervised learning.
>
> 4. We extend our framework to dynamic settings with carryover effects via a hierarchical DP/RL approach, which lies beyond the scope of [2].
>
> * **Q1: Incorrect sieve representation.**
>
> 1. When prior knowledge about $f$ (e.g., smoothness) is available, one can use standard basis expansions such as polynomials, splines, or Fourier series, which provide valid approximations for smooth function classes.
>
> 2. When such knowledge is unavailable, potential misspecification from an imperfect sieve can be mitigated in two ways: (i) consider multiple candidate bases and use cross-validation to select the one with the smallest approximation error; (ii) tune the hyperparameter $\eta$, which controls the degree of misspecification. Larger $\eta$ yields a more conservative upper bound and greater robustness to approximation error.
>
> * **Q2: Value Function Estimation Error and Behavior Policy Exploration**
>
> - **Q2.1 How does its estimation error affect the performance of the optimal treatment output?**
>
>      We assess sensitivity to value function estimation error by training four fully connected networks with architectures $(128,64,32)$, $(256,128,64,32)$, $(256,128,64,32,16)$, and $(512,256,128,64,32)$ under a common pipeline (batch normalization, SiLU, dropout), yielding $R^2$ values of $0.6789$, $0.9055$, $0.8961$, and $0.9394$ against the ground truth value function for the last day.
>
>      We then evaluate the downstream [MSE of the ATE estimator](https://www.dropbox.com/scl/fi/k1lx1lh4bfmtd0xoqudox/DP_MSE_bar_CI_facets_by_model_without.pdf?rlkey=6tiekieuswhm5xwrfzb2p72eu&st=h4jnu8i9&dl=0). Our method, as well as NRD, remains stable across architectures; notably, even with moderate accuracy ($R^2=0.6789$), our method outperforms baseline designs.
>
> - **Q2.2 How to ensure $\pi_b$ to sufficiently explore the representation space?**
>
>      We ensure sufficient exploration via two strategies. First, $\pi_b$ is designed to span a wide range of imbalance levels (see our response to Reviewer GhJw (Q2)). Second, given $\pi_b$, we generate $B$ independent synthetic rollouts. As $B$ increases, the dataset covers a richer set of $(\Delta_n,\Gamma_n)$. In experiments, we set $B$ on the order of thousands to ensure adequate exploration.
>
> - **Q2.3 Justify the use of neural networks**
>
>     From a theoretical perspective, neural networks are justified by the smoothness of the value functions and classical approximation results [3]. In our setting, $Q_N$ in Eq. (9) is smooth in $(\Delta_N,\Gamma_N)$ and this property propagates to all $Q_n$ via backward induction. With state dimension $p+L$, standard ReLU results imply that $Q_n$ can be learned at a nonparametric rate depending on smoothness; e.g., for $d$-H$\ddot{o}$lder $Q_N$, the rate is $O_p(N^{-2d/(2d+p+L)})$ up to log factors [4].
>
>      Empirically, our results (see Q2.1) show that neural networks achieve high approximation accuracy with moderate model size.
>
> * **Q3: Issues on independence across the days**
>
>     The independence assumption is motivated by our primary application in ridesharing. Due to space constraints, we refer the reader to our response to Reviewer gneX (Q4) for details.
>
> [1]Near-optimal ab testing.
>
> [2]Sequential bayesian optimal experimental design via approximate dynamic programming.
>
> [3]Neural networks for optimal approximation of smooth and analytic functions.
>
> [4]Nonparametric regression using deep neural networks with relu activation function.

---

> > ### Author Rebuttal · Reviewer_j6i9 · 2026-04-04
> >
> > I appreciate the authors for addressing my concerns.

---

> > > ### Author Response · Authors · 2026-04-04
> > >
> > > Thank you for your positive feedback and for acknowledging that our response addressed your concerns. We sincerely appreciate it.

---

### Official Review · Reviewer_gneX · 2026-03-12

**Soundness:** 3
**Presentation:** 3
**Significance:** 3
**Originality:** 3
**Overall Recommendation:** 4
**Confidence:** 4

**Summary:**

This paper proposes a robust framework for sequential experimental design in both contextual bandits and dynamic settings. With features $X_n$ arriving sequentially at each step $n=1,\dots,N$, the goal is to  assign the treatment $a_n$ so that after collecting the data $(X_n,A_n,Y_n)_{n=1}^N$, the MSE of the OLS estimate for the ATE (under a working additive linear model) is minimized. Assuming an additive model with constant treatment effect and potential nonlinear covariate-dependent bias, the authors derive the formula of the MSE. To address the challenge that the MSE depends on the unknown nonlinear bias, the authors propose to use a sieve approximation and address its high dimensionality by minimizing an upper bound on the MSE. The authors show that this problem reduces to a "Bellman recursion" that can be solved by synthetically estimating the value functions and applying a dynamic programming. The framework is then extended to settings with interactive treatment effects and dynamic settings with time-varying treatments within a time horizon. The efficacy of the proposed methods are demonstrated on simulations under misspecified settings and semi-synthetic experiments.

**Compliance With Llm Reviewing Policy:**

Affirmed.

**Key Questions For Authors:**

1. (important for my evaluation) Although the authors claim robustness to model misspecification, the model (2) is still assuming a constant conditional treatment effect, which places strong assumptions on the potential outcomes. The model (11) in Section 3.4 is also not fully general. What would happen for the most general case with $f(X_i,A_i)$ instead of $f(X_i)$ in (2) and (11)?
2. How would the results change if the errors $\{e_i\}$'s are heteroskedastic, i.e., $\text{Var}(e_i\mid X_i,A_i) = \sigma(X,A)$ for some unknown function $\sigma(\cdot)$?
3. (important for my evaluation) The derivation of conditional MSE conditions on all the $N$ pairs of $(a,X)$. Is this still correct if the treatment $a_n$ depends on past observations (outcomes)? Does the proposed treatment assignment involve $Y_t$'s?
4. The assumptions in Section 4 are a bit unclear. Do you mean $X_{it}$'s are independent across $i$, while dependent across $t$ for a fixed $i$? If one considers the dynamic setting (or RL), it seems reasonable to consider $X_{it}$ to depend on earlier history, too, thinking of $Y_{it}$ as the "rewards" and $X_{it}$ as the "states". Could you clarify how these assumptions might make sense in applications or how these assumptions have been used in the literature? Also please formalize the conditions on the random variables so it is possible to contextualize the results in across-day DP and within-day DP later on.
5. (important for my evaluation) Apart from MSE estimation, an important task in A/B testing is statistical inference, i.e., constructing confidence intervals for treatment effects. Does the current framework allow asymptotically normal confidence interval construction?
6. (important for my evaluation) The proposed framework minimizes an upper bound on MSE, as a proxy, to minimize the MSE robustly. Do you have any ideas on how close these two quantities might be, i.e., how informative this upper bound minimization can be?

**Limitations:**

yes

**Strengths And Weaknesses:**

Strengths:

1. The studied problem (experimental design) is important.
2. The paper seems to contribute a new method to the literature which might be of interest to the community.
3. The idea of robustifying the MSE minimization is interesting.
4. The experimental results are promising.

Weakness:

1. The results seem to still rely on assumptions of additive effects, etc, so it's unclear how general the framework is (see my questions).
2. It is unclear whether the proposed methods allow statistical inference.
3. I have some technical questions regarding some theoretical derivations (see my questions).

---

> ### Author Rebuttal · Authors · 2026-03-31
>
> Thank you for the valuable comments.
> - **Q1 & W1: Generality of the framework.**
>   Our framework extends to the case $f(X,A)$. Due to space limitations, we defer details to Reviewer Km6o (Q1).
> - **Q2 & W2: Extension to heteroskedastic errors.**
>   Our proposal can be extended to the heteroskedastic case, but the optimization is more challenging. We adapt the robust design of [1] to handle it, assuming $\text{Var}(e_i\mid A_i,X_i)=\sigma^2 g_i$ with unknown $g_i$. We replace OLS with a weighted least squares estimator using $\mathbf{W}=\text{diag}(w_1,\dots,w_N)$. The MSE structure is similar to Eq. (3), with the first term involving $g_i$, and the second term involving $f$ handled as before using $\mathbf{W}^{1/2}\mathbf{Z}$. To account for unknown $g_i$, we minimize the worst-case MSE over $\{g_i\}$ satisfying $\sum_{i=1}^N g_i^2\le N$. The first term is bounded as
>   $$I_1^W =\sigma^2\sum_{i=1}^N g_i w_i^2(\mathbf{u}^\top(\mathbf{Z}^\top\mathbf{W}\mathbf{Z})^{-1}Z_i)^2 \le \sigma^2\sqrt{N}(\sum_{i=1}^Nw_i^4(\mathbf{u}^\top(\mathbf{Z}^\top\mathbf{W}\mathbf{Z})^{-1}Z_i)^4)^{1/2}$$
> by the Cauchy-Schwarz inequality. The term $\mathbf{u}^\top(\mathbf{Z}^\top\mathbf{W}\mathbf{Z})^{-1}Z_i$ depends on $A_i$ and $\sum_{i=1}^N w_i A_i$. This introduces fourth-order terms into the objective, making optimization more complex. The applicability of DP is uncertain and we leave the optimization to future work.
> - **Q3 & W3: Derivation of conditional MSE.**
>   Our conditional MSE is derived given a fixed treatment vector $\mathbf{a}$ and design matrix $\mathbf{X}$. It is valid as long as each $a_i$ depends only on covariates and past treatments (not outcomes), which our design satisfies. In the Appendix (Theorem 1), we optimize the expected upper bound of the MSE over treatment rules where $a_i$ is $\mathcal{F}_i$-measurable, with $\mathcal{F}_i$ including past covariates and treatments but excluding responses. Thus, the conditional MSE is well-defined. If treatment depends on past outcomes, the conditional MSE is ill-defined due to treatment revealing residual information. This issue does not arise in our setting, but the conditional MSE can still serve as an asymptotic proxy for the unconditional MSE under additional conditions.
> - **Q4: Assumptions in dynamic settings.**
>   Yes, we assume independence across days and temporal dependence within each day. Specifically, for day $i$, $(X_{i,t+1},Y_{i,t})$ may depend on the history $(X_{i,1:t},A_{i,1:t},Y_{i,1:t-1})$. For simplicity, we adopt a Markov assumption, which is commonly used in RL. This assumption is well motivated in applications such as ridesharing, where very low demand in early morning hours (e.g., 1am--5am) effectively resets the system, making each day approximately independent (see Fig.1 in [2]). While cross-day dependence may exist, it is typically much weaker than within-day dependence, and similar assumptions are common in the A/B testing literature [2,3]. We will clarify this point if accepted.
> - **Q5: Does the current framework allow asymptotically normal CI construction?**
>   The ATE estimator is asymptotically normal but may be biased under misspecification. Nevertheless, when the bias is moderate, it can be ignored. Then standard Wald-type procedures may be used. We conduct simulations to evaluate the performance of the CI for the setting in Section 5.1. [Table 1](https://www.dropbox.com/scl/fi/l0xmi9b5qok3qeteanw3f/tab-1.png?rlkey=gslmqs5h9leldwpd5fjl5mhzx&st=umvwuy89&dl=0) suggests that our design achieves near-nominal coverage. For comparison, we also report results under a random design, which undercovers the true ATE. These results suggest that the standard CI performs well empirically. Developing theoretically valid CI is a vital direction for future work but is beyond the scope of this paper.
> - **Q6: How informative can this upper-bound minimization be?**
>   We address this question in two cases.
>   1. **Case 1: Correct specification.** The bias vanishes, so MSE equals variance and the gap to our upper bound is the worst-case bias. Following [1], we conducted simulations to show the efficiency loss for robustness. Due to space limitations, please refer to our first response to Reviewer Km6o.
>   2. **Case 2: Misspecification.** The upper bound is sharp in a worst-case sense. For a given treatment vector $\mathbf{a}$, if $f(X)=\eta \mathbf{\Psi}(\mathbf{X})\mathbf{U}\boldsymbol{\kappa}$ with $\boldsymbol{\kappa}^\top \propto \mathbf{a}^\top\mathbf{\Psi}(\mathbf{X})\mathbf{U}$ and $\lVert \boldsymbol{\kappa} \rVert_2=1$, then the Cauchy–Schwarz inequality is tight, and the proposed upper bound is asymptotically equivalent to the true MSE; see Prop A.1 and Eq. (A.3).
>
> [1] Integer-valued, minimax robust designs for estimation and extrapolation in heteroscedastic, approximately linear models.
>
> [2] Policy evaluation for temporal and/or spatial dependent experiments.
>
> [3] Trustworthy online marketplace experimentation with budget-split design.

---

> > ### Author Rebuttal · Reviewer_gneX · 2026-04-01
> >
> > Thanks for the response! Re Q3: Why does the treatment only depend on past treatment and covariates? I thought the Q-learning etc uses the response information.

---

> > > ### Author Response · Authors · 2026-04-02
> > >
> > > We sincerely appreciate the reviewer’s feedback. Below, we address this concern in more detail.
> > >
> > > Our treatment assignment depends only on past treatments and covariates because we focus on the design problem that minimizes the mean squared error (MSE) of the ATE estimator, rather than directly optimizing the responses themselves as in standard Q-learning. In standard Q-learning, the response is explicitly treated as the reward signal, and the goal is to maximize the cumulative reward. In contrast, under our setup, the ``reward'' is defined as the negative MSE, which depends primarily on the bias and variance of the ATE estimator—not on the realized responses directly.
> > >
> > > More specifically, as shown in Eq. (9), the MSE objective depends on the outcomes only through the noise variance $\sigma^2$ and the misspecification level $\eta$. These quantities are fixed and can be calibrated using historical data when available. Similarly, in [1], the treatment assignment also does not depend on the responses, reflecting the same design principle.
> > >
> > > [1] Near-optimal ab testing.

---

### Decision · Program_Chairs · 2026-04-30

**Decision:**

Accept (regular)

**Comment:**

This paper introduces a dynamic programming approach to experimental design for A/B testing that targets a worst-case upper bound to the MSE on the treatment effect coefficient.

All reviewers backed the acceptance of the paper, praising its relevance and technical soundness.  There are some concerns about the novelty of the work (especially compared to Bhat et al, but as the authors explain in their rebuttal, they are (amongst other differences) importantly allowing for misspecification in the output's dependency on the covariants that previous work has not.  I believe that this provides a sufficient distinction for publication at ICML.  There are also concerns about the tightness of the bounds being optimised and the restrictiveness of the assumptions being made.  While I do believe these concerns have some merit, the authors do a good job of addressing them in their rebuttal and I think the work is still useful in spite of potential limitations on this front.

Overall, this is solid work which I believe meets the bar required for ICML and I recommend that it is accepted.